# Coordinated wound responses in a regenerative animal-algal holobiont

Dania Nanes Sarfati [1], Yuan Xue[2], Eun Sun Song[3], Ashley Byrne [4], Daniel Le[4], Spyros Darmanis[4], Stephen R. Quake[2,3], Adrien Burlacot [1,5], James Sikes [6] ✉ & Bo Wang [2,7] ✉

Animal regeneration involves coordinated responses across cell types throughout the animal body. In endosymbiotic animals, whether and how symbionts react to host injury and how cellular responses are integrated across species remain unexplored. Here, we study the acoel *Convolutriloba longifissura*, which hosts symbiotic *Tetraselmis* sp. green algae and can regenerate entire bodies from tissue fragments. We show that animal injury causes a decline in the photosynthetic efficiency of the symbiotic algae, alongside two distinct, sequential waves of transcriptional responses in acoel and algal cells. The initial algal response is characterized by the upregulation of a cohort of photosynthesis-related genes, though photosynthesis is not necessary for regeneration. A conserved animal transcription factor, *runt*, is induced after injury and required for acoel regeneration. Knockdown of *Cl-runt* dampens transcriptional responses in both species and further reduces algal photosynthetic efficiency post-injury. Our results suggest that the holobiont functions as an integrated unit of biological organization by coordinating molecular networks across species through the *runt*-dependent animal regeneration program.

The concept that multicellular organisms can only exist in conjunction with their symbionts has profoundly transformed our understanding of a variety of biological processes. The term "holobiont" has been introduced to refer to the collection of a multicellular organism and its microbial symbionts. This term carries an important implication: the host and its symbionts must function together as an integrated unit of biological organization[1,2]. Supporting this, studies on holobionts have revealed impacts of multispecies integration on physiological, behavioral, evolutionary, and developmental processes[3,4]. There has been a particular focus on how symbionts influence host development, such as *Vibrio fischeri* guiding the formation of the squid's light organ[5,6] and *Blochmannia* rewiring Hox genes during the embryonic development of Camponotini ants[7]. Conversely, the effects of the host on its symbionts

during critical developmental processes, which could significantly affect the holobiont as a whole, is of equal importance but has been less explored. Tissue regeneration is one such critical event. Regeneration is widely observed in the animal kingdom, though varies greatly among species[8,9]. Studies on animals, such as axolotl, zebrafish, planarian, and hydra, have demonstrated that regeneration involves intricate coordination across different cell types[10–12]. Considering animals as holobionts adds an additional layer of complexity as the regeneration program needs to be coordinated across species[13,14]. Understanding whether and how endosymbionts respond to the host regeneration and whether symbiont responses are modulated by the host regeneration program can unravel the mechanisms by which holobionts integrate molecular networks across species to operate as a single biological unit.

[1]Department of Biology, Stanford University, Stanford, CA, USA. [2]Department of Bioengineering, Stanford University, Stanford, CA, USA. [3]Department of Applied Physics, Stanford University, Stanford, CA, USA. [4]Chan Zuckerberg Biohub, San Francisco, CA, USA. [5]Department of Plant Biology, Carnegie Institution for Science, Stanford, CA, USA. [6]Department of Biology, University of San Francisco, San Francisco, CA, USA. [7]Department of Developmental Biology, Stanford University School of Medicine, Stanford, CA, USA. ✉e-mail: jsikes@usfca.edu; wangbo@stanford.edu

Progress in this respect has been bottlenecked by our limited capacity to measure the physiological states of symbionts. Here we overcome this challenge by studying animal-algal photosymbiosis, a common type of endosymbiotic relationship found in many regenerative animals, particularly in the groups of Porifera, Cnidaria, Acoela, and Mollusca[15]. In this relationship, the animal host benefits from the photosynthetic capabilities of algal partners, which fix carbon into organic compounds by converting solar energy to chemical energy through the photosynthetic electron transport chain[15]. Importantly, photosynthesis can serve as a gauge for algal physiology. In free living algae, photosynthesis is modulated by multiple abiotic factors including light[16], nutrients[17], temperature[18], and the availability of inorganic carbon[19]. In photosymbiosis, hosts can regulate the photosynthetic output of their endosymbiotic algae by modulating the concentrations of inorganic carbon and nitrogen[20,21], adjusting the pH of the symbionts' microenvironment[22,23], and changing light intensity[24]. The study of regeneration in photosymbiotic animals, though rarely explored[25], presents a great opportunity to unravel the molecular integration between host and symbionts.

Specifically, we study the photosymbiotic acoel *Convolutriloba longifissura*, a marine worm that maintains an obligate symbiosis with *Tetraselmis* green algae[26]. The algae reside between acoel cells, making this relationship an extracellular endosymbiosis[27]. The acoels acquire their symbionts after hatching, and can transfer the algae to their clonal progeny through asexual fission[28,29]. *C. longifissura* fissions every few days, generating new individuals through regeneration of its entire body from tissue fragments[28,30]. In contrast to embryonic development[31], regeneration proceeds in the presence of algal symbionts.

*C. longifissura* has been grossly understudied and the molecular tools available to investigate its biology are generally lacking. Therefore, we first assembled high-quality transcriptomes for both the acoel and the alga and developed a suite of tools to evaluate the host and endosymbionts' responses during regeneration at the molecular and physiological levels. We found that, along with the expected acoel responses, the algae exhibited an abrupt decrease in photosynthetic efficiency, accompanied by large scale transcriptional changes including the upregulation of pathways related to photosynthesis, carbon concentrating mechanisms, and chlorophyll biosynthesis during early phases of regeneration. Notably, this contrasts the transcriptional changes induced by light stress in a similar timeframe, implicating that host injury triggers algal responses different from common stress responses. These early responses were followed by a second, synchronized wave of transcriptional changes at later stages of regeneration in both the acoel and the algae. Knockdown of a conserved injury-induced acoel transcription factor, *Cl-runt*[32,33], blocked acoel regeneration, attenuated both the early and late transcriptional changes, and further decreased the operating yield of photosystem II (PSII). As the function of *Cl-runt* during acoel regeneration appears to be conserved, our results suggest that the acoel's early wound response contributes towards integrating the responses across species within this holobiont.

## Results

### Epimorphic anterior regeneration involves algal colonization

*C. longifissura* belongs to a derived family of acoels (Fig. 1a), the Convolutidae, which is the only family of the Acoela taxon that has evolved both endosymbiosis and robust asexual reproduction[34]. The animal's orange-red color is due to its red pigment cells and the green chloroplasts within the algal symbionts (Fig. 1b, c). The chloroplasts, autofluorescent in the red spectrum, facilitate fluorescence imaging to identify algal distribution during homeostasis and regeneration (Fig. 1d–f). Algal cells are distributed throughout the acoel's body, primarily accumulating beneath the body wall[27], which is a simple layer of epidermis lined with muscle and gland cells. A lower density of algal cells can also be found in the inner vacuolated parenchyma (Fig. 1d).

The anterior of the *C. longifissura* body contains the neural ganglion and two eye spots, while the posterior pole is characterized by a three-lobed tail. Two pairs of white concrement granules, situated laterally, are present on the acoel's dorsal surface: one posterior to the head and the second posterior to the gut syncytium[35] (Fig. 1b). The second pair of concrement granules coincides with the transverse fission plane[28], at which we bisected the animals to assess the wound responses and regeneration process.

The head fragments, undergoing posterior regeneration, did not form an obvious blastema. Newly formed tails regained their characteristic three-lobed morphology at 2 days post amputation (dpa, upper panels, Fig. 1f), similar to regeneration after fission[28]. Fluorescence imaging revealed that algal cells persisted in the posterior wound and regenerating tissues (upper panels, Fig. 1f). BrdU staining, which labels cells in S-phase and their progeny after division[30], indicated limited proliferation in the posterior wound region (Fig. 1g). These observations suggest that posterior regeneration follows a morphallactic process, wherein the preexisting tissues are remodeled to restore the original form of the animal with minimal cell proliferation[36].

In contrast, during anterior regeneration from tail fragments, at 1 dpa, BrdU$^+$ cells accumulated towards the wound site and formed a clear blastema (Fig. 1g), distinguished by its transparency due to the absence of pigment and algal cells (lower panels, Fig. 1f). Accumulation of cells in the blastema was observable with DAPI staining at 2 dpa (Fig. 1h). The new tissue expanded over the next couple of days, with head structures, including eye spots and the neural ganglion, restored by 4 dpa (Fig. 1f, h, Supplementary Fig. 1). These features of anterior regeneration are consistent with epimorphosis, where regeneration is mostly driven by cell proliferation[36].

The blastema was initially devoid of algae, which only repopulated new tissues between 3 and 4 dpa (lower panels, Fig. 1f). Live imaging revealed that algae are motile within the acoel tissue (Fig. 1i, Supplementary Movies 1, 2). Notably, at 2 dpa, algal cells appeared to be mostly restricted from the blastema, yet by 3 dpa, they freely moved in and out of the newly regenerated tissue. As the overall algal content was stable throughout the course of regeneration (Supplementary Fig. 2), this series of observations suggests that algae colonize new tissues primarily through random dispersal of algal cells from preexisting tissues during the late stages of regeneration. Given that anterior regeneration requires tissue growth and algal repopulation, we chose it as our focus for the rest of the study.

### Photosynthetic efficiency decreases upon acoel injury

Acoels regenerated normally when kept in the dark throughout the regeneration process (Fig. 2a). Animals treated with 3-(3,4-dichlorophenyl)−1, 1-dimethylurea (DCMU), a chemical inhibitor of PSII, showed impaired PSII activity (Supplementary Fig. 3) but similar regeneration rates as controls. Longer DCMU treatment eliminated algal cells and led to animal death (Supplementary Fig. 4). These observations suggest that algal photosynthesis is required for the long-term survival of the holobiont, but dispensable for regeneration.

We then tested if algae respond to acoel injury by measuring photosynthetic efficiency. For this, we designed a custom acoel chamber (Supplementary Fig. 3a, b) mounted on a Pulse-Amplitude-Modulation (PAM) fluorometer that allowed us to measure the photosynthetic changes of algae at a population level. At any moment, a PSII can be available to receive electrons (open center) or unavailable (closed center, occupied or damaged). When a PSII is open and exposed to light, it accepts photons and uses them for photochemistry. When a closed center receives light, the energy is transferred to an open center, dissipated as fluorescence or heat. By measuring chlorophyll fluorescence changes, we inferred the ratio of open and closed centers[37,38] (Fig. 2b). The maximum quantum yield of

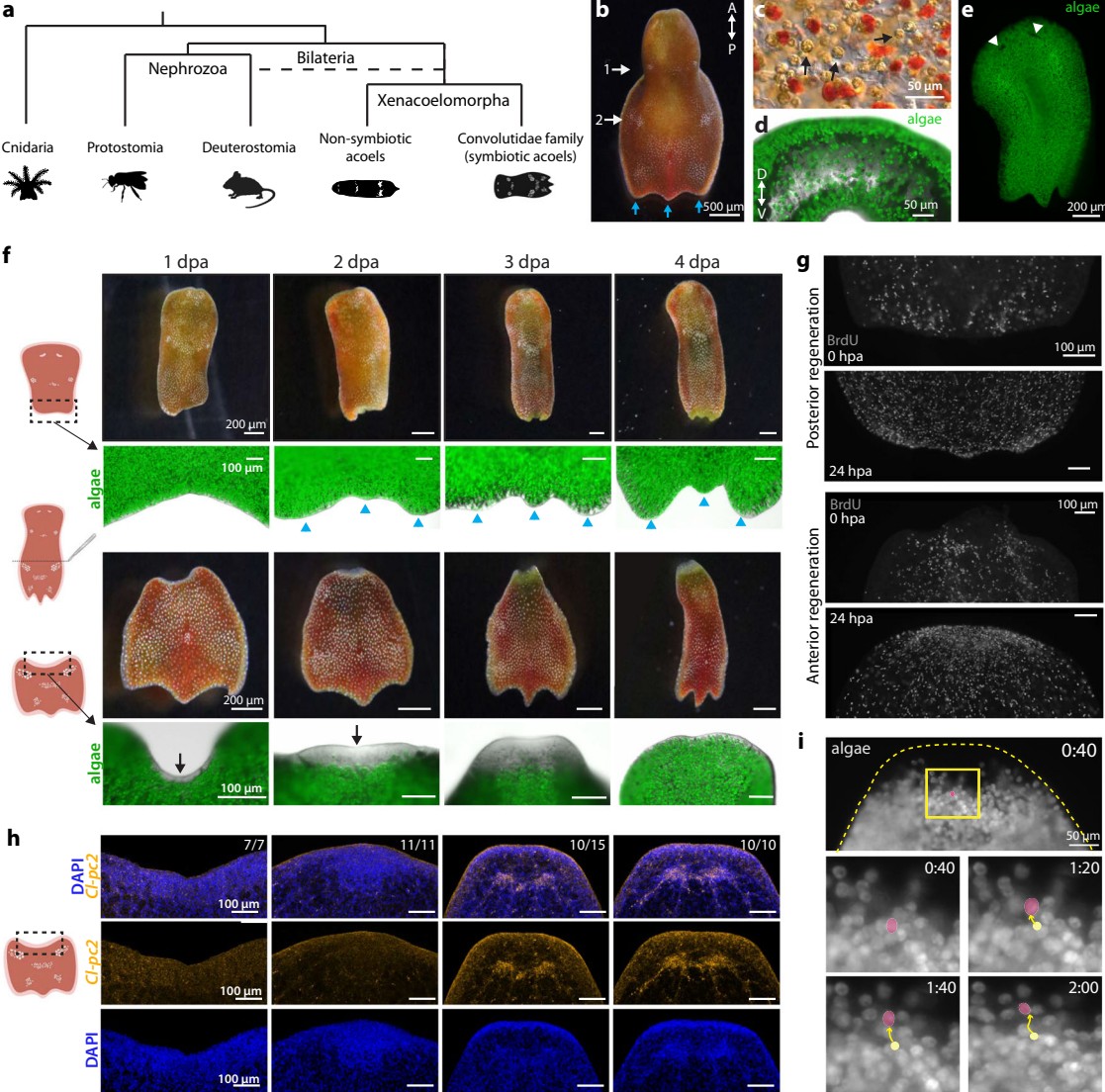

**Fig. 1 | Acoel regeneration involves both acoel and algal cells. a** Schematic of a simplified phylogeny showing the position of symbiotic acoels, which belong exclusively to the Convolutidae family. The position of the Xenacoelomorpha is still debated[92] as a basal bilaterian or a sister group to Ambulacrarians (dotted line). **b** Photograph of the acoel *C. longifissura*. Numbers mark the pairs of white concrement granules and blue arrows point to the three tail lobes. **c** Differential interference contrast image showing the acoel cells (transparent and red cells) intermingled with green-brown algal cells (a few examples are highlighted by arrows). **d** Transverse section at the second concrement granule pair showing the distribution of algae (green) along the animal dorsal ventral axis (D-V), imaged through autofluorescence at 647 nm. **e** Algal cells are ubiquitously distributed across the host body, except the eye spots (arrowheads), which are devoid of algae. **f** Regeneration from posterior and anterior facing wounds. Top: posterior regeneration, with blue arrowheads indicating the regenerated tail lobes. Bottom: anterior

regeneration with black arrows pointing at the blastema. Note the new tissue is devoid of algae until 4 dpa. **g** BrdU staining of posterior (top) and anterior (bottom) facing wound sites at 0 and 24 hpa. A buildup of BrdU⁺ cells is visible at the anterior wound at 24 hpa. **h** Fluorescence in situ hybridization (FISH) of the neuronal marker *prohormone convertase 2* (*Cl-pc2*) and nuclei staining with DAPI at various time points post-amputation. The blastema formation is evident by the accumulation of cells at 2 dpa based on the DAPI staining. Neural ganglia are regenerated by 3 dpa. Numbers represent animals in which the neural structure is consistent with the figure out of the total number of animals examined. **i** Snapshots showing algal motions at the boundary of anterior blastema at 3 dpa. Highlighted in magenta is an example of an algal cell moving towards the regenerated tissue. Yellow: original position. Time stamp is shown as minutes:seconds. The corresponding video is available as Supplementary Movie 2. Experiments were repeated at least twice (**c**, **d**, **g**, **h**) or ten times (**e**, **f**). Schematics are created with BioRender.com.

PSII ($F_v/F_m$) measures the maximum fraction of open centers after a dark incubation, allowing all undamaged centers to open, as a proxy for photo-inhibition. The quantum yield of PSII (Y(II)) measures the fraction of open centers at any given time with background light, depicting the efficiency with which light energy is converted into chemical energy[38,39]. If the photosynthetic electron transport chain gets saturated or compromised, PSII centers remain closed, reducing Y(II).

As a reference for comparison, we first determined the effects of exposure to high light, at an intensity of 150 $\mu$mol m⁻² s⁻¹, which is triple the light intensity to which the algae are acclimatized to and

the upper limit before the system reaches full saturation (Supplementary Fig. 3d). We quantified $F_v/F_m$ and Y(II) on tails (amputated and immediately measured), as whole animals moved too much to be measured reliably. We compared controls kept under constant light ("0 hpa", hours post amputation) and acoels exposed to high light for 24 h before measurement (labeled as "0 hpa + light stress") (Fig. 2c, d). As expected, high light exposure caused a decrease in $F_v/F_m$ which was paralleled with a decrease in PSII efficiency for all actinic light intensities tested (Fig. 2c, d).

We then evaluated the effects of amputation on photosynthesis in a matched timeframe (0 and 24 hpa). We added a 3 hpa time point as

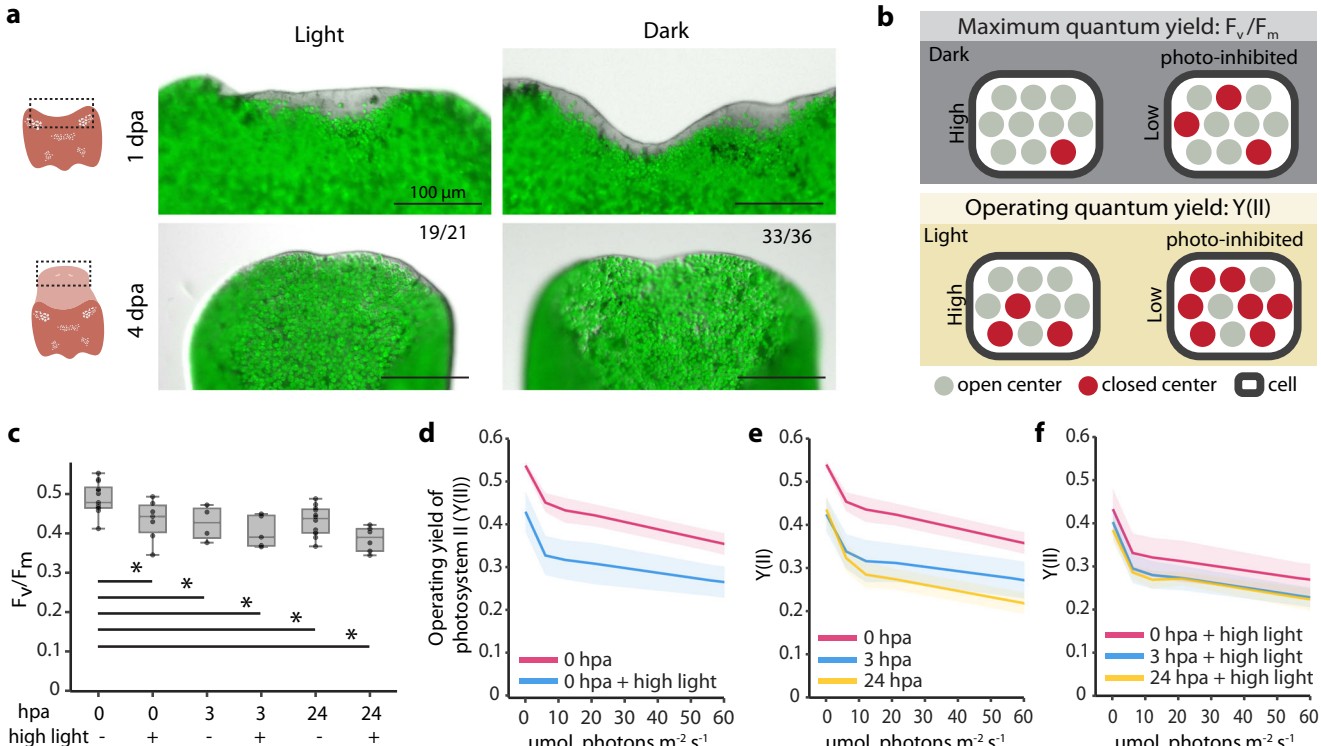

**Fig. 2 | Photosynthesis is not necessary for regeneration but is affected by amputation. a** Anterior regeneration is consistent in the presence or absence of light at 1 and 4 dpa. Algal cells (green) are imaged at 647 nm. Numbers represent animals which regenerated normally out of the total number of animals examined, experiments were repeated three times. Schematics are created with BioRender.com. **b** Schematics showing the biological interpretation of $F_v/F_m$ and Y(II). **c** Box plot showing $F_v/F_m$ of tails at different hpa, with or without light stress. For 0 hpa, samples were dark acclimated and then amputated under green light. Experiments contain 13 (0 hpa), 4 (3 hpa), 12 (24 hpa), 5 (3 hpa + light stress), 6 (0 hpa + light stress), and 7 (24 hpa + light stress) biological replicates, respectively. Boxes: upper and lower quartiles with the median marked by the middle line, whiskers: upper and lower fences, dots: individual replicates. All individual

treatments are significantly different compared to '0 hpa', with no significance between any other conditions (one-way ANOVA, $p = 6.53e{-}5$). *Tukey post hoc test, 0 hpa vs: 0 hpa + light stress, p-adj = 0.054; 3 hpa, p-adj = 0.083; 3 hpa + light stress, p-adj = 0.003; 24 hpa, p-adj = 0.001; 24 hpa + light stress, p-adj = 0.0096. Y(II) measurements at increasing actinic light intensities with and without light stress (**d**), at different time points post amputation without light stress (**e**), or exposed to 24 h high light before amputation (**f**). Experiments contain 3 (0 hpa), 4 (3 hpa), 5 (24 hpa), 7 (0 hpa + light stress), 3 (hpa + light stress), and 6 (24 hpa + light stress) biological replicates, respectively. Lines: mean; shaded regions: one standard deviation above or below. Each biological replicate consisted of 20 tails (**c–f**). Source data are provided as a Source Data file.

animal wound responses are expected to be already pronounced at this early time[12,32,33], which may also induce changes in algal physiology. Both $F_v/F_m$ and Y(II) decreased concordantly at 3 hpa (Fig. 2c) but only Y(II) continued to decrease at 24 hpa (Fig. 2e). The persistent decrease in Y(II) but not $F_v/F_m$ may be caused by further interference with the electron transport chain's photoconversion capacity at this later time point without affecting the maximum fraction of functional open centers. This suggests that host injury primarily induces a decrease in the sink capacity for photosynthates downstream of PSII, thereby limiting electron flow and eventually decreasing $F_v/F_m$. It is noteworthy that light stress reduces $F_v/F_m$ typically through a different mechanism: excess light overexcites the photosynthetic machinery, saturating the photosynthetic electron transport chain and producing reactive oxygen species (ROS), which result in degradation of photosystems in a process called photo-damage[40,41].

Light-stressed animals at 3 or 24 hpa showed barely reduced $F_v/F_m$ beyond the effect of light stress alone (Fig. 2c), while Y(II) had a minor decrease at 3 hpa without further reduction at 24 hpa (Fig. 2f). Both $F_v/F_m$ and Y(II) in amputated animals under light stress were comparable to values at 24 hpa without light stress, suggesting that the effects of high light exposure and acoel amputation are non-additive.

### De novo transcriptome assembly for the holobiont
Given that both acoel injury and high light exposure altered photosynthetic efficiency, we wondered how the molecular responses they

may induce differ. To characterize gene expression changes, we needed reference transcriptomes that include genes expressed during homeostasis and after injury, which were lacking for both *C. longifissura* and *Tetraselmis* sp. Therefore, we assembled de novo transcriptomes using tissues collected throughout an early regeneration time course (Fig. 3a, Supplementary Table 1). To obtain full length mRNAs and enhance the quality of the assembly, we used Nanopore long-read sequencing to generate reads covering larger portions of transcripts with greater overlaps between fragments, which can reduce ambiguity during assembly. However, Nanopore reads had frequent errors disrupting open reading frames (ORFs) of assembled transcripts. To address this, we polished the transcriptome using Pacbio ISO-Seq and Illumina short-read sequencing of the same cDNA (Supplementary Fig. 5a, b). After removing duplicate and chimeric contigs, we obtained a final transcriptome consisting of 21,191 transcripts including both acoel and algal genes.

To separate the transcripts based on species, we separately sequenced DNA from acoels treated with DCMU to eliminate algae (acoel-enriched sample), and flow-sorted algal cells (algal-enriched sample) (Fig. 3b, see methods). After aligning the reads to the transcriptome, we quantified the depth (the number of reads mapped to the transcript normalized by the transcript length) and coverage (the fraction of the transcript covered by sequencing reads) of each transcript in both samples (Fig. 3c, Supplementary Fig. 5c). Since the input for this experiment was genomic DNA (gDNA), transcripts encoded by

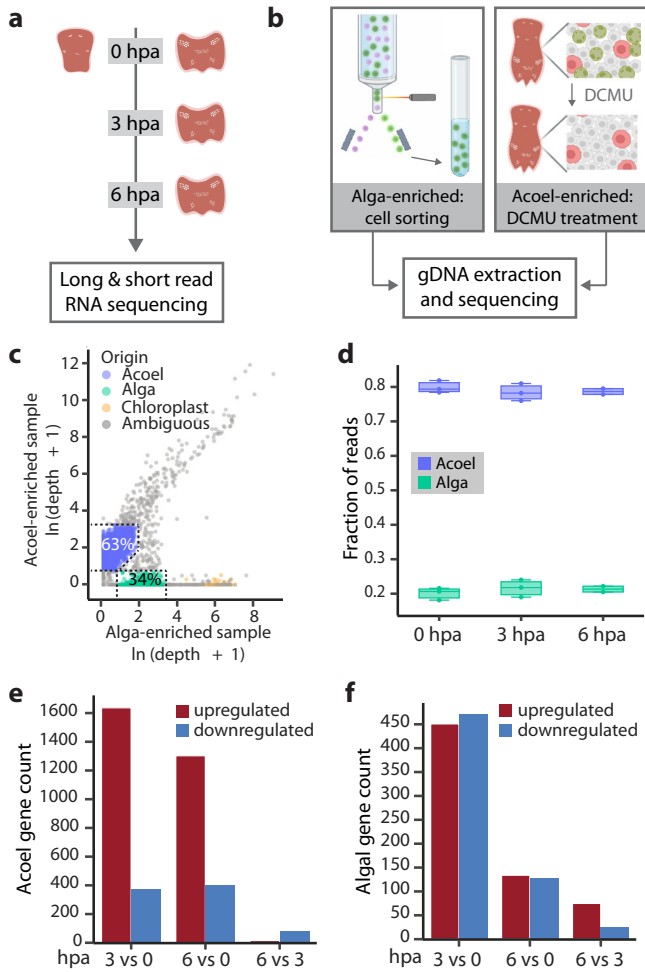

**Fig. 3 | Assembly of reference transcriptomes enables the measurement of molecular wound responses in both acoel and algal cells. a** Tails at 0, 3, and 6 hpa and heads at 0 hpa were collected for long and short read RNA sequencing to assemble reference transcriptomes de novo for both the acoel and the alga. Three replicates were obtained from 0 and 3 hpa tails, two from 6 hpa, and one from 0 hpa heads, with each replicate containing 5 animals. **b** Experimental design for identifying the species origin of transcripts through sequencing the DNA extracted from acoel-enriched and alga-enriched samples. **c** Sequencing depth of each transcript in the acoel-enriched and alga-enriched samples. Dotted lines indicate the gates used to assign acoel (blue) and algal (green) origin. The percentage of transcripts classified in each category is shown. Chloroplastic transcripts are annotated based on their high abundance in the alga-enriched sample and GO term predictions (cellular component GO term: 0009507 or 0009535, localized in chloroplast). **d** Fractions of acoel (blue) and algal (green) reads are consistent across samples and treatments, based on the Illumina short-read sequencing of tail samples specified in (**a**). Boxes represent upper and lower quartiles with the median marked by the middle line, the bars represent the upper and lower fences. Number of DEGs in acoel (**e**) and algae (**f**) at different time points post amputation (log₂FoldChange ≧ 0.8 or ≦ −0.8, p-adj ≦ 0.05, calculated with DESEQ2 using Wald test followed by Benjamini and Hochberg correction). Red: upregulated genes; blue: downregulated genes. Schematics are created with BioRender.com.

the same genome should have similar sequencing depths and should be enriched in one species over the other. Based on this, we selected conservative gates to separate acoel and algal transcripts, which were also supported by the differential sequencing coverage and contrasting GC content of transcripts between the two species (Supplementary Fig. 5c, d). With this classification, we recovered 13,313 acoel transcripts and 7216 algal transcripts (Supplementary Data 1, 2). The species origin of a small number of transcripts (662) remained ambiguous (Fig. 3c), which may be encoded by high copy number genes within the

acoel or algal genomes, mitochondrial and chloroplast genomes, or from bacterial and viral contamination. Among these ambiguous transcripts with high depth in the algae-enriched sample, we annotated 54 putative chloroplastic genes based on the Gene Ontology (GO) term predictions (Fig. 3c) including two homologs of the large subunit of Rubisco and multiple subunits of PSI, PSII, and ATP synthase.

Finally, we annotated the transcriptomes through BLAST and Trinotate (Supplementary Fig. 5a, Supplementary Data 3). The resulting transcriptomes have BUSCO scores comparable to other species of acoels and algae (Supplementary Table 2). The average mapping rates of Illumina sequencing reads to these transcriptomes were ~95% for all our RNA sequencing (RNA-seq) experiments, allowing us to reliably compare gene expression across conditions.

To analyze gene expression in each symbiotic species, we normalized the fractional coverage of acoel and algal transcripts separately. While the read count ratio of acoel to algal transcripts remained consistent after injury (Fig. 3d), we identified hundreds of differentially expressed genes (DEGs) in both acoel and algae at 3 hpa compared to 0 hpa, and these responses mostly persisted at 6 hpa (Fig. 3e, f). This result demonstrates that algae respond to host injury with large transcriptional changes in a time frame matching the host molecular wound responses.

**Upregulation of photosynthesis-related genes post-injury**

To identify algal pathways that respond to host amputation, we conducted a GO term analysis on the differentially expressed algal transcripts. Multiple terms related to photosynthesis, as well as chlorophyll and carotenoid biosynthesis, were upregulated (false discovery rate, FDR ≦ 0.1) at 3 hpa (Fig. 4a). In contrast, genes related to cell division, cell motility, and carbohydrate metabolism were downregulated (FDR ≦ 0.1) in algae at the same time point, consistent with the observation that algae do not repopulate the wound site at these early regeneration stages.

Examining specific genes supporting the GO term enrichment, we found 19 genes in the photosynthetic pathway collectively upregulated at 3 hpa (p-adj ≦ 0.05) (Fig. 4b). This list includes several components of PSII, such as the accessory subunits *Ts-psb28* and *Ts-hcf136*, thought to be required for PSII de novo assembly and repair[42,43], and multiple homologs of *psbS*, reported to regulate light stress responses in other algae[44]. We found similar trends in enzymes responsible for stabilizing the reaction center in the oxygen evolving complex in plants and green algae, *Ts-psbP* and *Ts-psbQ*[45], and in components of the light harvesting complexes (LHC) of both PSI and PSII, which are essential for the capacity to absorb light energy[46] (Fig. 4b). A homolog of *light harvesting complex stress-related 3* (*Ts-lhcsr3-1*), involved in safe dissipation of excess light energy into heat in microalgae[47] was upregulated transiently at 3 hpa (Fig. 4b).

We also observed 21 genes involved in chlorophyll biosynthesis upregulated after host injury (Fig. 4c). Upregulation of components of the photosynthetic complexes and chlorophyll biosynthesis has been implicated as part of a priming program that triggers a tolerance response in stress resistance in plants[48–50]. The carotenoid biosynthesis pathway was also upregulated at 3 hpa. This pathway is known to produce pigments commonly used by a variety of phototrophic organisms for photoprotection[51], which implies that host injury induces a photoprotective or acclimation response in the algae.

During photosynthesis, electrons generated by photochemistry at PSII are transferred through the cytochrome b₆f to PSI, creating high energy electrons used to reduce ferredoxin (Fd, *petF*) (Fig. 4b). The ferredoxin:NADP(H) oxidoreductase (FNR, *petH*) then generates reduced NADPH for CO₂ fixation in the process known as linear electron flow. Both *Ts-petF* and *Ts-petH* were upregulated after injury (Fig. 4d). In addition, alternative electron flows using reduced Fd have been described, including cyclic electron flow (CEF) and pseudo-cyclic

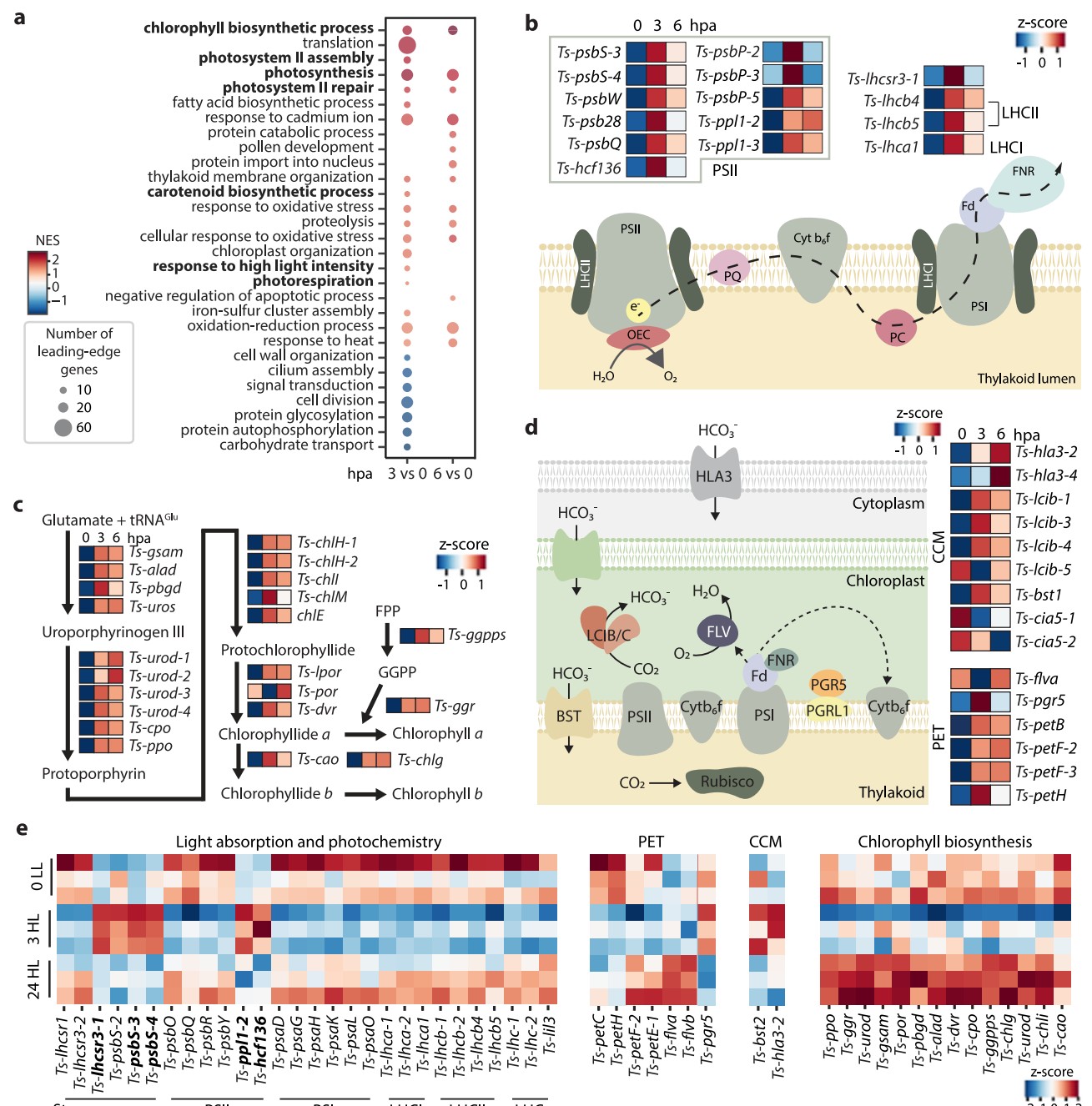

**Fig. 4 | Host injury induces distinct molecular responses in algae. a** GO term analysis of algal responses. Bolded GO terms are related to photosynthesis, chlorophyll and carotenoid biosynthesis pathways. NES: normalized enrichment scores. Diagram of the photosynthetic pathway (**b**), chlorophyll biosynthesis (**c**), carbon concentrating mechanisms (CCM), and alternative electron flow machinery (**d**), displaying DEGs (p-adj ≦ 0.05, calculated with DESEQ2 using Wald test followed by Benjamini and Hochberg correction). Heatmaps report z-scores of average expression at 0, 3, and 6 hpa measured from three biological replicates, each containing 5 acoels. LHCII light harvesting complex of PSII, LHCI light harvesting complex of PSI, e⁻ electron, PQ plastoquinone, Cyt b$_6$f cytochrome b$_6$f, PC plastocyanin, FPP farnesyl pyrophosphate, GGPP geranylgeranyl pyrophosphate. **e** Heatmaps showing normalized expression of genes associated with light absorption and photochemistry, photosynthetic electron transport (PET), CCM, and chlorophyll biosynthesis under low light (0 LL, 50 μmol m⁻² s⁻¹), after 3 h exposure to high light (3 HL, 150 μmol m⁻² s⁻¹), and 24 h exposure to high light (24 HL). Each row represents a biological replicate containing 5 acoels. Bolded transcripts are affected similarly at 3 HL and 3 hpa.

electron flow (PCEF)[52]. During CEF, electrons are transported from PSI through Fd and cytochrome b$_6$f back to PSI, and this activity is regulated by PGR5 and PGRL1[53]. The *Ts-petB* subunit of cytochrome b$_6$f, and *Ts-pgr5* were upregulated after injury (Fig. 4d). During PCEF, the electrons are used by flavodiiron proteins for the conversion of oxygen to water[54]. After injury, we also observed an upregulation of one of the two flavodiiron genes (*Ts-flva*) at 3 hpa. These results are in contrast with the reduction of photosynthetic electron transport capacity upon

injury as measured by a decrease in F$_v$/F$_m$ and Y(II)), suggesting that algae cells may compensate for loss of photosynthetic capacity with a higher transcription of these components.

In order to provide sufficient CO$_2$ to the CO$_2$-fixing enzyme Rubisco, microalgae have evolved CO$_2$-concentrating mechanisms (CCM) that transport and concentrate inorganic carbon (CO$_2$, HCO$_3$⁻) inside the chloroplast for its assimilation by Rubisco[52]. After amputation, three homologs of the low CO$_2$-inducible proteins b (*Ts-lcib-3, −4, −5*)[55,56],

were upregulated at both 3 and 6 hpa (Fig. 4c), along with bicarbonate transporters of the plasma membrane (*Ts-hla3-2, −4*)[57] and thylakoid membrane (*Ts-bst1*)[58] (Fig. 4d). We also detected downregulation of the transcription factors *Ts-cia5-1* and *Ts-cia5-2*, which regulate $CO_2$ responsive genes[59]. These observations indicate that the inorganic assimilation pathway is modified upon amputation, potentially reflecting a decrease in inorganic carbon availability.

Since upregulation of genes involved in photoprotective mechanisms like *lhcsr* or *psbS* are recurrent in microalgae subject to a high light stress[40], we compared molecular responses to light stress and host injury. Upon high light stress, most genes involved in light absorption and photochemical reactions were downregulated acutely in response to high light exposure at 3 h (labeled as "3 HL") and returned to the baseline by 24 h of high light exposure ("24 HL") (Fig. 4e). This is in concordance with previous studies in plants and free-living microalgae under light stress[40,60]. Only a handful of genes responded similarly at 3 h after either light stress or amputation, including a subset of *Ts-psbS, Ts-ppl1-2, Ts-lhcsr3-1*, and *Ts-hcf-136* homologs, which potentially represent generic stress responses as they are also induced by various stresses like $CO_2$ limitation[61] and UV exposure[40] in other green microalgae.

We noted a similar difference in the chlorophyll biosynthesis pathway, with most upregulated genes at 3 hpa being downregulated in 3 HL samples, with some becoming upregulated in 24 HL (Fig. 4e). Genes involved in photosynthetic electron flow and CCM exhibited more complex patterns under high light, which did not overlap with injury-induced responses (Fig. 4e). These observations, combined with the decrease in photosynthetic efficiency (Fig. 2c–f), suggest that although both light and injury converge on affecting algal photosynthesis, their molecular responses diverge.

### A *runt* transcription factor is essential for regeneration

To determine if algal responses are coupled with the host regeneration program, we first focused on identifying regulators of acoel regeneration in order to perturb the host regeneration program and evaluate how it affects the algal responses. We found several acoel candidate regulators upregulated at 3 or 6 hpa (Supplementary Fig. 6a). These included a set of conserved RNA binding proteins often associated with multipotency in diverse organisms, such as homologs of *vasa, piwi*, and *argonaute 2*[62], along with conserved transcription factors (TFs), such as *egr, runt, fosl*, and *klf* homologs, some of which have been reported to play important roles in regeneration of other animals[32,33,63]. To further narrow down the list, we compared DEGs at 3 hpa between *C. longifissura* and a non-symbiotic acoel, *Hofstenia miamia*, which has been studied for its regeneration capabilities[32]. We found only a few genes differentially expressed in both species at this time point (Fig. 5a). Of these shared DEGs, two were transcription factors: *Cl-runt* and *Cl-egr* (Fig. 5a), both of which had a single homolog in our transcriptome.

We validated the induction of *Cl-egr* and *Cl-runt* after injury in *C. longifissura* using in situ hybridization (Fig. 5b, Supplementary Fig. 6b, c). While *Cl-egr* was activated in both anterior and posterior wounds, *Cl-runt* expression was specific to anterior wounds, clearly demonstrating the difference between the anterior and posterior regeneration programs. We proceeded to knock down *Cl-runt* and *Cl-egr* via RNA interference (RNAi). Although all *Cl-runt* RNAi animals closed the wound, none of the tail fragments formed a clear blastema or regenerated a head, evidenced by the absence of the neural ganglion at 4 dpa along with other head structures (Fig. 5c, d), whereas only ~15% of *Cl-runt* RNAi treated anterior fragments failed to regenerate a tail (Supplementary Fig. 6d). Knockdown of *Cl-egr* led to regeneration deficiencies at a lower penetrance – only a third of either heads or tails failed to regenerate (Supplementary Fig. 6e, f). The highly reproducible anterior regeneration phenotype of *Cl-runt* RNAi was used to evaluate whether and how the algal response to injury depends on the host.

### *Cl-runt* RNAi alters algal response to host injury

To determine whether the algal transcriptional responses to host injury depend on the regeneration program controlled by *Cl-runt*, we performed RNA-seq on animals at 0 and 3 hpa after *Cl-runt* RNAi (Supplementary Fig. 7a, b). Strikingly, we identified a large number of DEGs not only in the acoel, but also in algae, suggesting that *Cl-runt* influences the transcriptional responses in both partners, likely through intermediate signaling molecules or other factors in the symbiont.

We noticed that RNAi treatment significantly reduced *Cl-runt* expression, but did not entirely eliminate it after host injury (Fig. 6a). This enabled us to calculate the correlation between *Cl-runt* expression and the expression of all other genes, providing an alternative analysis to identify genes modulated by *Cl-runt* RNAi in both acoel and algae.

We identified algal genes that may be activated by the *Cl-runt*-mediated injury response by selecting genes that were either significantly downregulated in *Cl-runt* RNAi animals compared to control at 3 hpa (p-adj $\leqq$ 0.1, $\log_2FC \geqq 0.8$) or in strong positive correlation with *Cl-runt* expression ($\rho \geqq 0.8$). Of the 91 genes selected, 20 are related to light harvesting and photochemical reactions, of which 8 are LHC proteins and 12 are subunits of PSI and PSII (Fig. 6b, c). Many of these genes continued to respond to injury in the *Cl-runt* knockdowns, but their activation was dampened. Only three of these genes (*Ts-psbs-4, Ts-bst-2*, and *Ts-pgr5*) were differentially expressed at 0 hpa between the control and *runt* RNAi animals, suggesting that *Cl-runt* influences the photosynthesis pathway mostly during regeneration.

Genes involved in photosynthetic electron transport, including a cytochrome $b_6f$ subunit, plastocyanin *Ts-petE*, ferredoxin *Ts-petF*, FNR *Ts-petH*, and multiple PSI and PSII subunits, were correlated with *Cl-runt* expression, indicating that *Cl-runt* expression may contribute to sustain the photosynthetic electron transport in algae after host injury. To test whether physiological changes also depend on *Cl-runt*, we measured the photosynthetic efficiency after *Cl-runt* RNAi. While the knockdown of *Cl-runt* did not affect $F_v/F_m$ at either 0 or 24 hpa (Fig. 6d), Y(II) was significantly reduced at 24 hpa compared to the control (Fig. 6e, Supplementary Fig. 7c). These results imply that, while other injury-induced factors may be responsible for the reduction of the algal maximal photosynthetic capacity, likely via degradation of PSII, the *Cl-runt*-dependent responses contribute to sustaining the electron transport flow after injury.

We also observed multiple transporters associated with the *Cl-runt*-mediated response in both acoel and algae. In the algae, we found 7 transporters dependent on *Cl-runt* expression, including a bicarbonate transporter (*Ts-hla3-2*), two *zinc-iron permeases* (*Ts-zip3* and *Ts-zip12*), two sodium solute transporters (*Ts-slc5a7* and *Ts-slc6a13*), and a *urea active transporter* (*Ts-dur3*) (Supplementary Fig. 7d). In the acoel, we found 14 transporters affected by *Cl-runt* knockdown. Two of these, *Cl-slco4c1* and *Cl-vha*, were downregulated upon injury and could regulate the acidity of the extracellular environment[23,24,64]. A glutamate transporter, *Cl-eaat1*, was also downregulated after injury and could affect the nitrogen cycling between the symbionts. Multiple solute carriers were upregulated after injury, including *Cl-slc18b1, Cl-slc17a5, Cl-slc6a18, Cl-slc6a8*, and *Cl-slc6a5*, which may modify the amino acid, protein, and nutrient exchange with the algae during regeneration[65] (Supplementary Fig. 7e). This suggests that *Cl-runt* is likely involved in mediating the acoel-alga communication during regeneration, by tuning the exchange of metabolites and small molecules between the two organisms.

### Molecular responses remain coupled later in regeneration

To determine whether the algal responses extend beyond early time points post injury, we conducted RNA-seq at 0, 1, and 2 dpa. Compared to 0 dpa, at 1 dpa, we observed minimal changes in the expression of both acoel and algal genes, though *Cl-runt* and *Cl-egr* were upregulated

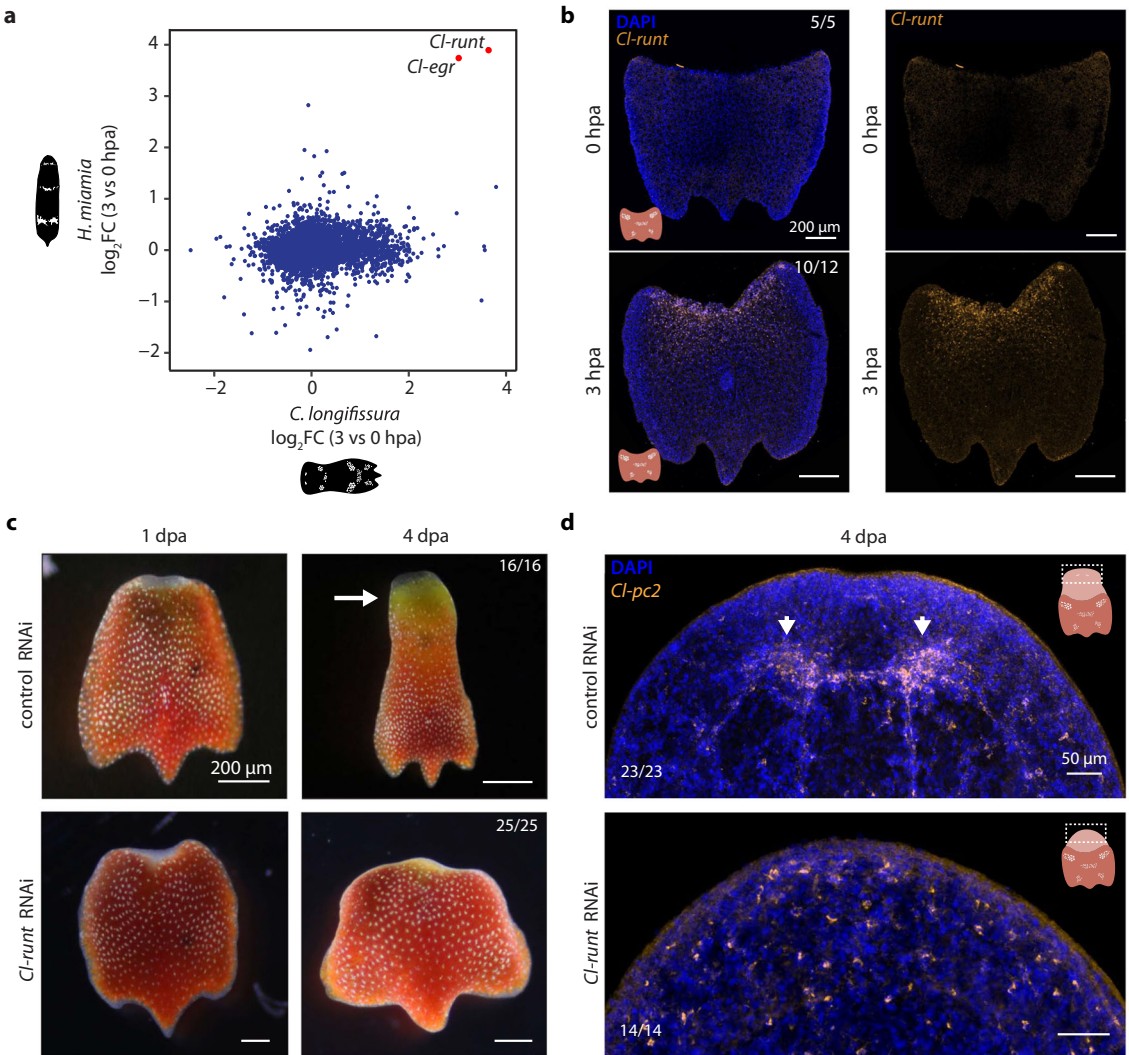

**Fig. 5 | *Cl-runt* transcription factor is a conserved regulator of acoel regeneration. a** Comparison of log2FoldChange (log2FC) in expression of orthologous genes at 3 vs 0 hpa between *C. longifissura* and *H. miamia*. Orthologs were identified using reciprocal BLAST. Differentially expressed genes at 3 hpa for *H. miamia* were obtained from ref. 32. **b** Expression of *Cl-runt* is evident only at 3 hpa at anterior wounds. Numbers represent animals in which the expression of *Cl-runt* is consistent with the figure, out of the total number of animals examined. **c** Morphological comparison between *Cl-runt* and control RNAi treated animals at 1 and 4 dpa. While *Cl*-runt RNAi tail fragments heal the wound, they neither form a blastema at 1 dpa nor regenerate head structures and eye spots, and have minimal tissue growth and shape changes. Arrow points at the regenerated head. **d** Expression of *Cl-pc2*, a neuronal marker, at 4 dpa in control and *Cl-runt* RNAi treated acoels. Neuronal ganglion is only present in control animals, whereas only distributed neurons are observed in *Cl-runt* RNAi treated animals. Arrowheads point at the regenerated lobes of the neuronal ganglion in the control RNAi animal. Numbers in (**c**, **d**) represent the animals that had regeneration phenotypes consistent with the images shown out of the total animals examined. Experiments were repeated twice (**b**, **d**) or three times (**c**) with similar results. Source data for (**a**) are provided as a Source Data file.

at this time point (Supplementary Fig. 8a, b). In contrast, hundreds of genes were differentially expressed at 2 dpa (Fig. 7a, b, Supplementary Fig. 8c, d). Importantly, there was little overlap between algal genes responding at earlier and later time points (Fig. 7c, d). For example, the responses within the photosynthetic electron flow pathway seemed exclusive to the algae's early responses, whereas a set of TFs, mostly squamosa binding proteins[66], were upregulated solely in the late response (Fig. 7d, e). We also noticed that the algal transporters differentially regulated at early or late time points were different, suggesting that the exchange mechanisms between acoel and algae may shift during regeneration. These observations imply that, by 2 dpa, the acoel and algae had jointly switched to a second wave of transcriptional responses. Consistently, knockdown of *Cl-runt* abolished this second wave of responses in both species (Fig. 7a, b, e), demonstrating that the algal responses are coupled with the progression of the host regeneration.

## Discussion

In this study, we established the acoel *C. longifissura* as a system to dissect the molecular integration across symbiotic partners during whole-body regeneration, using a suite of sequencing techniques and functional genomic analysis. We demonstrated that in addition to the animal's regeneration responses, the endosymbiotic algae also respond transcriptionally and physiologically to the host injury. Several lines of evidence suggest that the algal responses are tightly coupled with the host regeneration program. First, both the acoel and algae exhibit two distinct waves of transcriptional responses – at early and late stages of regeneration, respectively – with a synchronized shift between these waves around 1 dpa. Second, algae undergo a specific response to acoel injury characterized by a decrease in algal photosynthetic efficiency shortly after host injury, accompanied by the upregulation of genes involved in the photosynthetic electron transport chain. Finally, these transcriptional responses in both species are

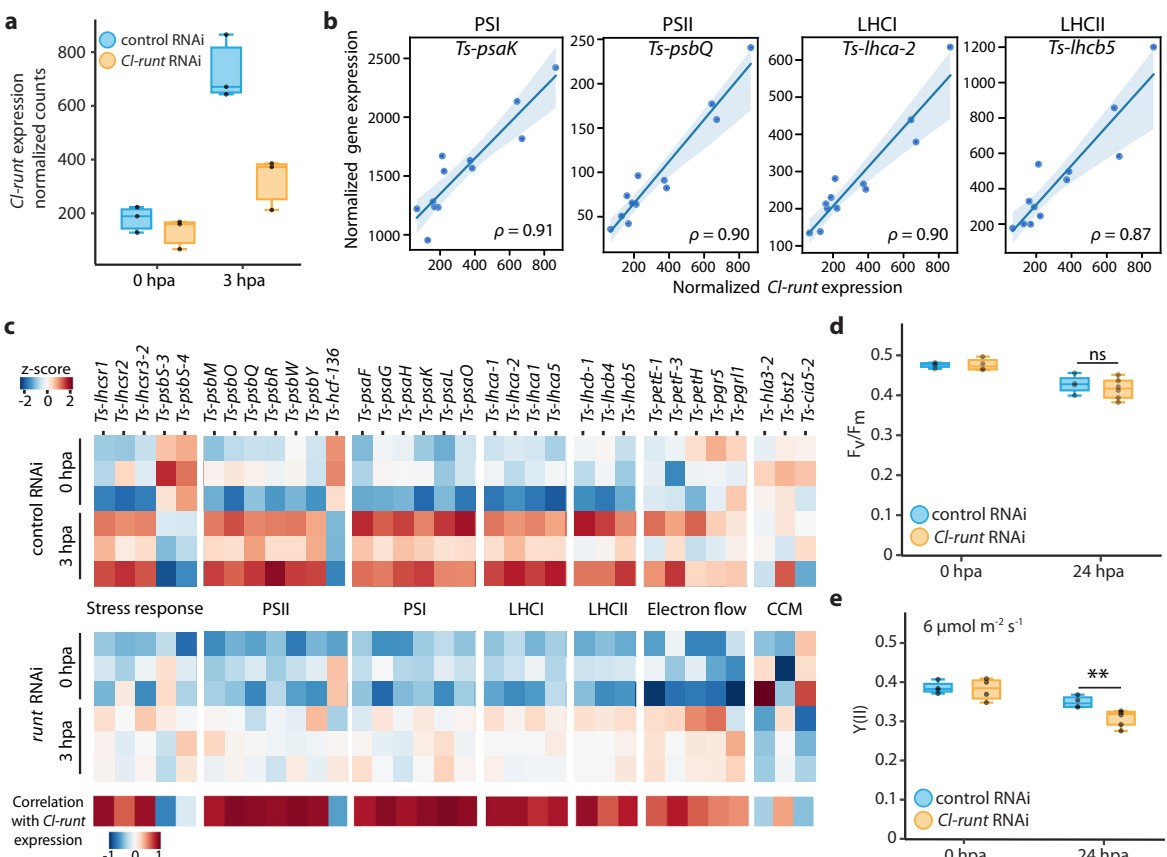

**Fig. 6 | *Cl-runt* affects transcription of photosynthetic related genes and photosynthetic efficiency. a** Expression of *Cl-runt* at 0 and 3 hpa in control (blue) and *Cl-runt* (yellow) RNAi animals, with three replicates shown by dots. **b** Normalized expression of select photosynthesis genes vs *Cl-runt* normalized counts. Each dot represents a sample from the RNA-seq experiment. Replicates are shown as dots, lines represent the linear regression, and shaded regions are the confidence interval of the regression. ρ: Spearman correlation. The examples are chosen to represent components of PSI, PSII, LHCI, LHCII, and PET, respectively. **c** Left: heatmap reporting z-scores from triplicates of selected genes involved in the light reactions of photosynthesis, alternative electron flows, and CCM that are differentially expressed at 3 hpa in *Cl-runt* RNAi vs control RNAi, or highly correlated with *Cl-runt*

expression. Bottom: correlation of the given gene with *Cl-runt* expression. **d** $F_v/F_m$ measured at 0 and 24 hpa in *Cl-runt* RNAi and control tails. There is a similar decrease in both *Cl-runt* RNAi and controls after 24 hpa. **e** Y(II) of *Cl-runt* RNAi and control tails at 0 and 24 hpa with 6 μmol photon m⁻² s⁻¹ of actinic background light. Both *Cl-runt* RNAi and controls show a decrease in Y(II) at 24 hpa compared to their counterparts at 0 hpa. Y(II) of *Cl-runt* RNAi at 24 hpa compared to control RNAi is significantly reduced (**$p = 0.011$, ns not significant, two tailed *t*-test). For (**d, e**) each treatment consisted of four biological replicates, and each replicate contained twenty tails. Boxes represent upper and lower quartiles with the median marked by the middle line, the bars represent the upper and lower fences. Source data are provided as a Source Data file.

modulated by *Cl-runt*, a host TF induced by injury. Knockdown of *Cl-runt* halts the regeneration process and nearly eliminates the second wave of responses. In the early time points, *Cl-runt* RNAi attenuates the activation of the photosynthesis-related genes and further reduces the quantum yield of PSII, presumably due to a slower repair of damaged components of the electron transport chain. Together, these findings suggest that the molecular networks of the two species are linked through a regulatory program modulated by *Cl-runt* (Fig. 7f). This provides the molecular foundation that enables the holobiont to function as an integrated biological unit throughout regeneration.

Impaired photosynthesis is a common characteristic in general algal stress responses. However, our observations reveal that the molecular responses to injury diverge from those induced by light stress. This divergence is marked by the upregulation of many photosynthesis-related genes after injury that are instead downregulated under intense light. Intriguingly, the upregulation of some of these photosynthetic genes and pigment-related genes, such as *psbP*, *psbO, psbQ, urod*, has been described in stress-tolerant plants and tolerance primed plants[48–50]. Collectively, these observations suggest that, contrary to stress responses aimed at containing photo-damage, the injury-induced transcriptional responses may serve to compensate for the loss in photosynthesis caused by host injury.

We also propose that *runt* may be a conserved master regulator of acoel regeneration, given the comparison with the molecular responses of the non-symbiotic acoel, *H. miamia*[32]. It is plausible that injury-induced *runt* activation has been co-opted to activate algal responses. However, there is minimal overlap in wound responses between the two acoels, raising the possibility that the endosymbiotic algae may have modified the regeneration programs in *C. longifissura*, as endosymbiosis in acoels is a derived trait whereas regeneration is shared by multiple acoel clades. Analysis of regeneration in more acoel species will help to test this hypothesis.

As an animal transcription factor, it is unlikely that *Cl-runt* could directly control transcription in algae. While it remains unclear how the host and the symbiont cells communicate, we noted that several transporters appear to be regulated by *Cl-runt* in both animals and algae across the early and late stages of regeneration (Fig. 7e, Supplementary Fig. 7d, e), implying that the exchange of metabolites or signaling molecules may be involved in this process. Indeed, nutrients, nitrogen-based compounds, and organic and inorganic carbon are regularly exchanged between endosymbiotic partners during homeostasis, which may be modified during stress in other holobionts[20,67–70]. This exchange may lead to modified pathways that allow for coordinated regeneration responses in both the host and symbionts.

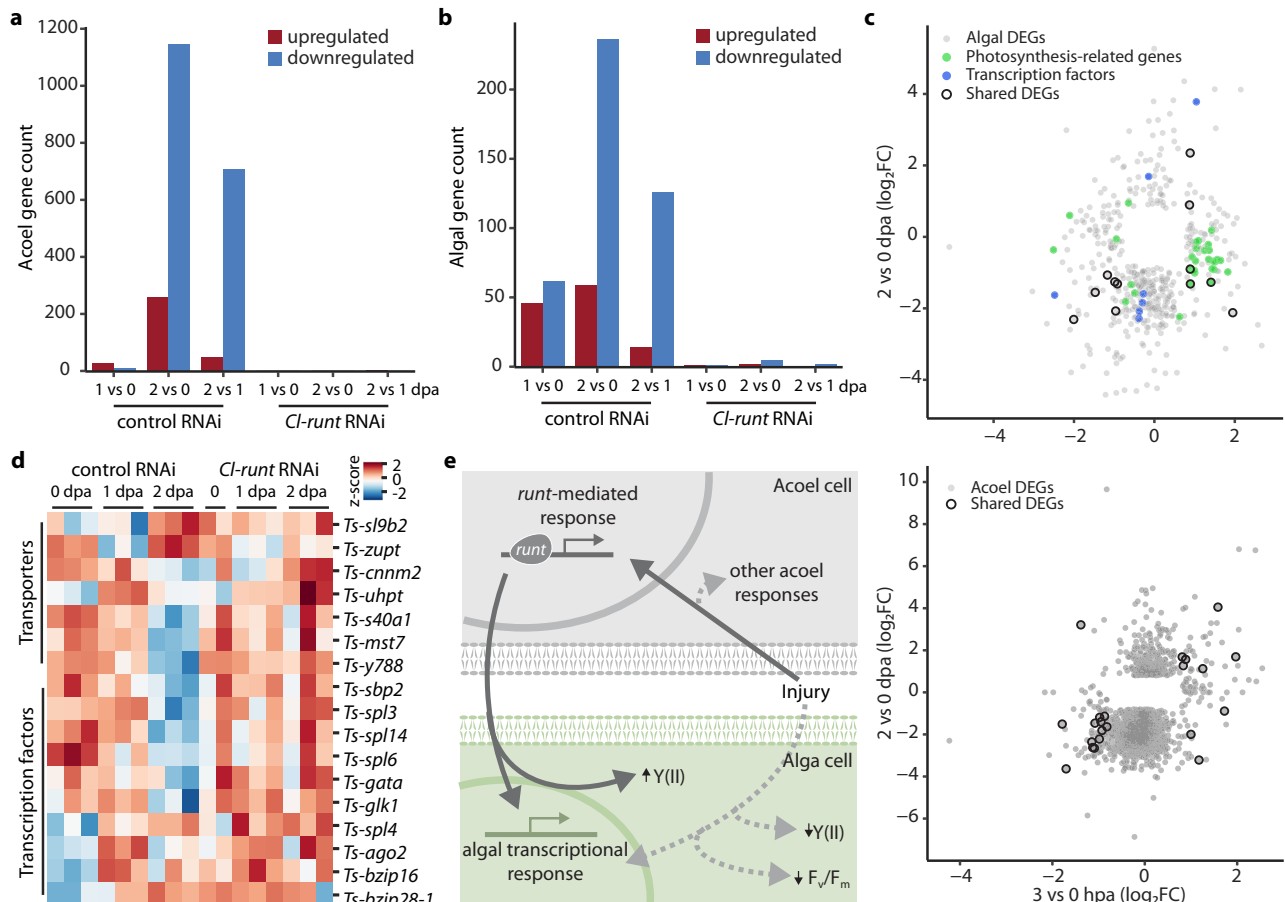

**Fig. 7 | Acoel and algal responses are coupled at late stages of regeneration.** Number of DEGs in acoel (**a**) and algae (**b**) at 1 and 2 dpa (log2FoldChange $\geq 0.8$ or $\leq -0.8$ for upregulated and downregulated genes, respectively, p-adj $\leq 0.05$; calculated with DESEQ2 using Wald test followed by Benjamini and Hochberg correction), in control and *Cl-runt* RNAi treated animals, which have drastically reduced numbers of DEGs. **c** Comparison of log2FoldChange (log2FC) for DEGs at 3 hpa and 2 dpa for algae (top) and acoel (bottom), showing minimum overlap between the early and late responses. Photosynthesis related genes, highlighted in green, are mostly only upregulated at 3 hpa. A set of algal TFs are differentially regulated only at 2 dpa, highlighted in blue. Black circles: genes differentially expressed at both time points ($n = 12$ for algae, $n = 22$ for acoel). **d** Heatmap

reporting z-scores from triplicates of selected algal genes at 1 and 2 dpa. All genes shown were found differentially expressed (p-adj $\leq 0.05$) between 2 and 0 dpa control RNAi samples or between *Cl-runt* and control RNAi samples at 2 dpa. The TFs correspond to those highlighted in blue in (**c**). **e** Schematic of proposed *Cl-runt*-mediated coordination between acoel and algal responses during regeneration. Dark gray arrows indicate responses dependent on *Cl-runt* activation. Activation of algal responses that are dependent on *Cl-runt* should be regulated indirectly through intercellular communication. The light gray, dashed arrows represent alternative pathways that may be activated during regeneration and influence algal responses in a *Cl-runt*-independent manner.

Two candidate communication pathways between host and symbionts are regulated by nitrogen and carbon metabolites. It has been reported that nitrogen availability is important for photosynthesis and proliferation of algae and can serve as a mechanism for the host to regulate algal endosymbiont proliferation in other systems[20,71]. We observed a set of genes involved in nitrogen cycling differentially regulated through regeneration. For example, at the early time points, the algal urea transporter *Ts-dur3* was upregulated, while an acoel H(+)/nitrate transporter (*Cl-sialin*-1) was upregulated and the acoel's *glutamate synthase* (*Cl-gogat-1*), *glutamate transporter* (*Cl-eaat1*), and *ammonium transporter 1* (*Cl-amt1-1*) were downregulated after amputation in a *Cl-runt* dependent manner (Supplementary Fig. 7d, e). Interestingly, the response often switched between homologs at early and late stages of regeneration. For example, we observed different homologs of *gogat*, (*Cl-gogat-2*, downregulated), *sialin* (*Cl-sialin-2*, downregulated) and *amt* (*Cl-amt1-2*, upregulate) responding at 2 dpa. In addition, carbon exchange could also play a role during regeneration. The availability of inorganic carbon is required for algal carbon fixation while the heterotrophic host requires organic carbon compounds for survival[22,23,52,67]. Recent studies suggest that in addition to the algal CCM, animal hosts also concentrate carbon for their algal

symbionts[22,23] with V-type H+-ATPase (VHA) acidifying the microenvironment and promoting the conversion of $HCO_3^-$ into available $CO_2$ for the algae. In *C. longifissura*, *Cl-vha* and *Cl-slco4c1*, organic anion transporters that may use $HCO_3^-$ as a counterion[64], were negatively correlated with *Cl-runt* expression and downregulated after injury (Supplementary Fig. 7e). On the algal side, the CCM master regulator *Ts-cia5-2* and the bicarbonate transporter *Ts-hla3-2* were negatively correlated to *Cl-runt* early during regeneration (Fig. 6c). By regulating carbon and nitrogen availability, these transporters could provide means to trigger the algal response to injury.

Algae may also recognize host injury without the need for direct communication by sensing common regeneration-induced physiological changes, such as accumulation of reactive oxygen species (ROS) and ascorbic acid, an antioxidant that scavenges ROS[72]. While elevated ROS levels have been noted during tissue regeneration in several animals[73,74], in photosynthetic organisms, ROS can induce transcriptomic responses that regulate most of the genes involved in light harvesting and photosynthetic electron flow[75,76]. In *C. longifissura*, *Cl-runt* activation leads to an increase in the expression of *gulonolactone oxidase* (*Cl-gulo*), which is responsible for ascorbic acid biosynthesis in animals[77]. Given that this gene is only affected at the early time points

post amputation, changes in both ROS and ascorbic acid concentrations may be sensed by the algae to activate their early responses specifically.

It is worth mentioning that other factors, independent of *Cl-runt*, should also contribute to the algal responses, as algae continued to have some physiological and transcriptional responses to host injury after *Cl-runt* RNAi, especially at early time points. These may include microenvironment changes caused by host injury, such as fluctuations in osmolarity, pH, and even light penetration, as well as other acoel molecular wound responses controlled by separate pathways.

Finally, although algal photosynthesis responds to host injury, we found that algal photosynthesis is not required for acoel regeneration. This finding aligns with observations in other photosymbiotic animals[25] and may reflect the resilience of these systems, which likely build on several alternative energy sources. Besides glucose, photosynthetic algae can produce lipids for energy storage, sustaining their long-term growth and survival[78]. *Tetraselmis* algae stand out for their lipid synthesis efficiency[79]. It is conceivable that the holobiont taps into these lipid reserves for the energy required during regeneration. In addition, the acoel may directly consume algae, as recently observed for corals and their photosynthetic symbionts[80]. Supporting this, we consistently observe the emergence of a feeding spot during regeneration[28,29] (Supplementary Fig. 1).

The interplay between hosts and endosymbionts during animal development has become increasingly recognized for its significance in the evolution and function of multicellular life forms[7,81]. Our approach offers a broadly applicable strategy to study the regulatory mechanisms by which animal hosts influence symbionts' responses during critical developmental processes such as regeneration. By blocking host development and analyzing the symbionts' response, we can determine whether symbionts contribute to the host's developmental processes, and how they adjust their own physiology to ensure their survival in the changing environment within the host tissue. Many photosymbiotic animals, including *C. longifissura*, develop in the absence of their symbionts and gain their symbionts in later stages of their life cycle[82]. Evaluating the differences between embryonic development and regeneration, as well as comparing regeneration between symbiotic and non-symbiotic species, could help to understand how gene networks are integrated across symbiotic partners in holobionts and how these integrations may have emerged over evolution.

## Methods

### Animal maintenance
Acoels were cultured in a 16 gallon Coralife Biocube Aquarium in artificial seawater (ASW, 34 ppt, Instant Ocean) at 26 °C with a 14 h:10 h light:dark cycle. Acoels were fed freshly hatched *Artemia* shrimp every 1–3 days *ad libitum* in the tank. Animals used in experiments were kept in individual wells in an incubator at 26 °C with a 14:10 h light:dark cycle, with filtered ASW. These animals were fed and their water was changed every 1–3 days. The light intensity to which the animals were exposed was modulated for individual experiments between 50 and 150 μmol photon $m^{-2} s^{-1}$.

### Regeneration experiments
Animals were kept individually or in groups smaller than 10 individuals in 12-well or 24-well plates in an incubator (Danby Fresh 1.7 cu. ft. Herb Grower, cat. #DFG17A1B). Pictures of animals were taken at the same time every day, over 4 days after amputation, using a stereo microscope (Zeiss Stemi 508) or an inverted fluorescence microscope (Zeiss Axio Observer Z1) to evaluate acoel and algae growth. For dark treatment, a chamber was used to keep acoels in the dark at 26 °C. Animals were not exposed to light before imaging. Individual animals were only imaged once. Dark treatment was replicated three times with three animals per replicate.

### Live imaging of algal motility within the acoel tissue
Acoels were anesthetized with 3% $MgCl_2$ in ASW and placed on a 35 mm Petri dish with a glass bottom. The animals were imaged on an inverted epi-fluorescence microscope (Zeiss Axio Observer Z1) through a 5× objective using both chlorophyll autofluorescence and bright field. Images were captured every 4 s for 5 min and then processed using ImageJ.

### Quantifying algal content using flow cytometry
Animals were dissociated on ice in the dissociation media (3.3× calcium magnesium free PBS, 2% FBS, 20 mM HEPES) by gently pipetting until solution was homogenized. The suspension was then filtered through a 40 μm strainer to remove debris and placed on ice. Cells were stained with 5 μM of Dye Cycle Violet (Invitrogen, cat. #V35002) for 20 min at room temperature. Before sorting, the solution was filtered again through a 35 μm strainer and gently mixed. On a cell sorter (Sony SH800S), singlets were first gated, and then algal and acoels cells were identified based on the DNA content and algal autofluorescence (Supplementary Fig. 2a). Number of events classified as algae were divided by the sum of algae and acoel events to calculate the algal content. For each experiment, 10 animals were pooled and dissociated together. Statistical significance when comparing across conditions was calculated using one way ANOVA.

### DCMU treatment
Acoels were treated with 20 μM DCMU in ASW, by diluting the stock solution of 20 mM DCMU in ethanol. Vehicle controls were treated with ethanol diluted in ASW (1:1000). After 24 h, acoels were rinsed multiple times and maintained in fresh ASW. During the treatment, the animals were kept unfed. Effects of the DCMU treatment were confirmed with a Dual-PAM system, showing blockage of PSII (Supplementary Fig. 3c). For long-term DCMU treatments, animals were treated with DCMU twice, 3 days apart, and fed every other day. Algae were removed from the acoel 2 weeks after the two treatments, validated through imaging using the red fluorescence of the algae at 633 nm.

### Photosynthesis measurements
To measure chlorophyll fluorescence in the algae within the animal, we created a chamber for containing acoels on a Dual-PAM 100 (Walz) pulse amplitude modulation (PAM) fluorimeter using a small plastic Petri dish and a 2 mm thick silicone container (Supplementary Fig. 3a). We carved a 5 × 6 mm space to contain the animals. On the edges of the acoel chamber we increased the height to 4 mm so that the diode of the emitter would not touch the water. This device was then mounted on a PDMS base that fit the detector diode in order to keep the sample stable during measurements. The Dual-PAM system was used in a vertical configuration (Supplementary Fig. 3b). We used 20 tail fragments for each replicate in all experiments. This number was determined to give sufficient fluorescence signal allowing for reliable quantification of $F_v/F_m$ and Y(II). Animals were incubated in the dark prior to measuring $F_v/F_m$ to ensure that all functional PSII were in the open state, which is required for calculating their maximum quantum yield.

To measure chlorophyll fluorescence, detection pulses (10 μmol photon $m^{-2} s^{-1}$ blue light) were applied at a 100-Hz frequency. Basal fluorescence ($F_o$) was measured after a 20 min dark adaptation prior to the first saturating flash. Red saturating flashes (6000 μmol photon $m^{-2} s^{-1}$, 600 ms) were delivered to measure $F_m$ (in the dark) and $F_m{'}$ (in the light). PSII maximum yields ($F_v/F_m$) were calculated as $(F_m-F_o)/F_m$. The operating quantum yield of PSII (Y(II)) was calculated upon actinic illumination as $(F_m{'} - F_o)/F_m$. For light curves, animals were exposed to increasing light intensities, and for each light intensity, acoels were acclimated to the light for 2 min prior to a measurement of Y(II) in order to quantify steady state photosynthetic rates.

Freshly amputated tails were used as controls, as they undergo the same handling procedure as other experimental groups, ensuring a consistent baseline for comparison. This approach also ruled out potential differences in algal photosynthesis between anterior and posterior tissues. Amputations were completed quickly in under 5 min, and performed under green light following dark incubation to minimize perturbations to the system. To enhance reproducibility of our measurements, these experiments were repeated multiple times on different dates, but consistently at the same time of day, to ensure that the results were not influenced by unaddressed variables.

In determining the time points post amupuation for measuring photosynthetic activity, we chose 24 hpa because, at this time, $F_v/F_m$ and Y(II) showed more pronounced responses compared to 3 hpa (Fig. 2). A synchronization of transcriptional and functional responses is not expected due to the time needed to translate changes in mRNA abundance into observable functional outcomes. Indeed, there was a small transcriptional response at 24 hpa, even though the changes in photosynthesis were most evident.

## RNA extractions

Three replicates each containing five animals were used for bulk RNA extraction under each condition. We rinsed the animals with ASW twice and then removed as much ASW as possible. 300 µL of Trizol was added, followed by a 2–3 min incubation at room temperature and vigorous vortexing to dissociate the tissue. Samples were then flash frozen on dry ice and kept at −80 °C until extraction so that all samples for each RNA-seq experiment could be processed in parallel. On the day of the extraction, samples were thawed on ice and 700 µL of fresh Trizol was added. After brief vortexing, 200 µL of chloroform was added and the sample was shaken vigorously for 15 s followed by a 2 min incubation. Samples were then centrifuged (16,000 $g$, 15 min at 4 °C) and the aqueous phase was carefully transferred into new tubes and processed with the Direct-zol RNA Purification Kit (Zymo Research, cat. #R2051) following the manufacturer's instructions, which includes a DNAse treatment step. RNA concentration and quality was quantified on a Bionanalyzer.

## RNA-seq library preparation

50 ng of input RNA was reverse transcribed (RT) using the Smartscribe Reverse Transcriptase (Clonetech). Full-length cDNA was generated using a modified Smart-seq2 protocol[83]. During the RT reaction a template-switch oligo and a custom oligo(dT) primer containing a UMI and sample barcode were supplemented to enrich mRNA (Supplementary Data 4). The RT reaction was performed in 10 µL reactions and incubated at 42 °C for 1 h. After first strand cDNA synthesis, 1 µL of 1:10 dilutions of RNAse A (Thermofisher) and Lambda Exonuclease (NEB) were added and incubated at 37 °C for 30 min. Following the incubation, an amplification step was performed using KAPA Hifi ReadyMix 2X (KapaBiosystems) containing 1 µL of ISPCR primer (10 µM) in 25 µL reactions. Samples were incubated at 95 °C for 3 min, followed by 12 PCR cycles of (98 °C for 20 s, 62 °C for 15 s, and 72 °C for 4 min), with a final extension at 72 °C for 5 min. Libraries were purified using a ratio of 1:0.85 sample to bead ratio using Agencourt AMPure XP SPRI beads (Beckman Coulter). The final products were quantified on a D5000 Tapestation (Agilent) or a Bioanalyzer using the High Sensitivity DNA kit (Agilent, cat. #5067-4626).

To generate sufficient amounts of input cDNA (>1 µg) for producing Oxford Nanopore Technologies (ONT) and PacBio libraries, which were used for the transcriptome assembly, samples were pooled together equally with 20 ng of cDNA each and amplified through an additional 6 PCR cycles. The entire pool was purified using Agencourt AMPure XP SPRI beads at a 1:0.85 ratio and eluted in 50 µL yielding a final library concentration of ~115 ng/µL.

For the ONT library preparation, ~1–2 µg of the final full-length cDNA product was prepared using the ligation based ONT method with the SQK-LSK109 kit, according to the ONT instructions with minor modifications. First, the end repair and A-tailing reactions were both extended to 30 min each at 20 °C and 65 °C, respectively, instead of 5 min for each reaction. Second, the ligation reaction time was extended to 30 min at room temperature instead of 10 min. Two runs were performed using the MinION device with the 48 h sequencing protocol in the MinION 9.4.1 chemistry flowcells. All bases were called with the high-accuracy GPU accelerated model of Guppy v3.5.2.

For the PacBio library preparation, a SMRT library was prepared with 1 µg of the full-length cDNA product using the Sequel II binding kit 2.0. Reads were processed through the Circular Consensus Sequencing (CCS) pipeline using Smartlink to generate high quality reads. Each read from the CCS output was generated using parameters of "min pass = 1" and "min accuracy = 0.85".

For Illumina sequencing, 4–10 ng of the full-length cDNA was used as input for preparing Nextera XT (Illumina, cat. # FC-131-1024) libraries following the manufacturer's recommendations. The input cDNA was indexed using a tagmentation reaction and then incubated at 72 °C for 3 min to heat inactivate the enzyme. The indexed cDNA libraries were amplified with 12 PCR cycles (95 °C for 10 s, 55 °C for 30 s, and 72 °C for 30 s), with a final extension at 72 °C for 5 min. Some amplified libraries were size selected for 300–800 bp on a 2% EX E-gel (Thermofisher) and purified using QIAquick gel extraction kit (Qiagen). Libraries were pooled at equal concentrations and sequenced either on a NextSeq 500 using High Output runs, or on a NovaSeq 6000.

## Transcriptome assembly and annotation

We filtered Illumina reads using Trimmomatic (v 0.39) with the following parameters "LEADING:3 TRAILING:3 SLIDINGWINDOW:4:15 MINLEN:36", and Nanopore and Pacbio reads using NanoFilt (v 2.7.1) with the following parameters "-q 10 -l 150 --headcrop 75 --tailcrop 75". We assembled an initial transcriptome draft with RATTLE assembler[84] using the filtered Nanopore reads. We then aligned filtered Illumina and Pacbio reads to the initial draft using Minimap2 (v 2.17-r941) to perform base polishing using Pilon (v 1.23)[85]. To remove chimeric transcripts or regions with poor read support, we performed a coverage scan with a rolling window size of 10 bp to trim both ends of the transcript that have less than 33% of the maximum alignment coverage. After trimming, transcripts that had lengths shorter than 300 bp were removed. We then re-mapped the Illumina reads to the initial draft using Salmon quant (v 1.3.0) and clustered similar transcripts with Grouper (https://github.com/COMBINE-lab/grouper) to remove duplicate transcripts. Illumina and Pacbio reads were then re-mapped against the clustered transcriptome using Minimap2 for a second round of base polishing, yielding the final version of the transcriptome.

We predicted the ORF for each transcript using TransDecoder (v 5.5.0) (https://github.com/TransDecoder/TransDecoder) with default parameters, only keeping the longest ORF. We performed functional annotation using Trinotate (v 3.2.1). To identify putative homologs, we performed blastx on the transcriptome against the NCBI "refseq_protein" database. To evaluate the completeness of our transcriptomes, we performed BUSCO (v 4.0.5) analysis.

## Determining the species origin of transcripts using DNA sequencing

We treated a large cohort of acoels with two rounds of DCMU, each lasting 24 h (see DCMU treatment), in order to remove algal cells and obtain an acoel-enriched sample. After 2 weeks of incubation, we confirmed the absence of algal cells through fluorescence imaging and selected animals with none or rare algal cells. We then proceeded to flash freeze these animals in liquid nitrogen and stored them at −80 °C. The alga-enriched sample was collected on a cell sorter (Sony SH800S) based on algal autofluorescence using the procedure described above. Sorted algal cells were centrifuged and washed twice with ASW before gDNA extraction.

gDNA was extracted using a high-molecular weight DNA isolation protocol[86] with some modifications. We used 400 µL of GTC buffer with a 30 min incubation for tissue dissociation, then added 200 µL of distilled water and 400 µL of phenol/chloroform, and mixed by inversion. We then centrifuged at 12,000 g for 15 min at 4 °C and transferred the upper aqueous phase into a new Phase Lock gel tube. 500 µL of chloroform were added, and the solution was mixed by inversion before centrifugation at 12,000 g for 10 min at 4 °C. The upper aqueous phase was again transferred to a new Phase Lock gel tube. We then added 200 µL of 5 M NaCl, mixed by inversion before incubating for 10 min on ice, and centrifuged for 10 min. The upper aqueous phase was then transferred to a DNA LoBind tube, 600 µL of cold isopropanol were added, and samples were stored at 4 °C overnight. Samples were centrifuged for 2 h at 4000 g at 4 °C. The pellet was washed with 1 mL of 70% ethanol and centrifuged at 4000 g for 10 min at 4 °C. The pellet was resuspended in 50 µL of TE buffer (10 mM Tris-HCl, 1 mM EDTA, pH 8.0). We quantified gDNA concentration on a Qubit. Libraries were prepared with the Illumina DNA Prep kit (cat. #20018704) following the manufacturer's instructions. 72 ng and 420 ng were used as input for the alga-enriched and the acoel-enriched samples, respectively. Reads were aligned to the transcriptome using Minimap2 (v 2.17-r941) with preset parameters for genomic short-read. We selected only properly paired mapped read counts for the analysis. Transcript depth and coverage was calculated with samtools (v 1.13, https://github.com/samtools/samtools).

### Differential gene expression analysis

Short reads were aligned with Minimap2 (v 2.17-r941), quantified with HTSeq, and differential gene expression analysis was performed using DESEQ2 (v 1.38.2), separately for the algal and acoel transcripts. For algal genes, the putative chloroplast genes were not included since our library preparation protocols enriched for polyadenylated transcripts and polyadenylation in chloroplast genes targets them for degradation instead of transcription[87]. GO term analysis was performed using the GSEAPy package (v 1.0.4)[88].

### BrdU staining

Acoels were exposed to 0.1 mg/ml BrdU for 2 h, washed, and amputated. At 0 or 24 hpa, animals were relaxed in 3.4% MgCl₂ for 5 min and then immediately incubated in 4% formaldehyde (FA) in ASW for 1.5 h. Samples were washed with PBS, denatured in HCl (3:1 in distilled water) at 37 °C for 30 min, washed with PBSTx (PBS supplemented with 0.1% Triton-X), and blocked in 10% goat serum in PBSTx (blocking solution) for 1 h, followed by an incubation with mouse anti-BrdU monoclonal antibody (Sigma cat. #B2531, 1:100 dilution in blocking solution) overnight at 4 °C. Samples were then washed multiple times with PBSTx before being incubated in FITC-conjugated goat anti-mouse secondary antibody (Sigma cat. #A6667, 1:200) for 2–5 h at room temperature. Finally, samples were mounted in 75% glycerol for imaging.

### In situ hybridization

Animals were rinsed with filtered ASW, relaxed briefly in 3.4% MgCl₂, and fixed with 4% FA in ASW for 1.5 h. Samples were then washed twice with PBSTw (PBS + 0.1% Tween-20) for 5 min each, followed by two quick rinses with 100% methanol. Fixed samples were stored in 100% methanol in −20 °C.

Riboprobes for in situ hybridization were synthesized using the oligonucleotide primers listed in Supplementary Data 4 to clone the DNA fragment of interest into vector pJC53.2 (Addgene Plasmid ID: 26536), followed by riboprobe synthesis as previously described[89].

Fixed animals were bleached in 6% H₂O₂ in methanol for 1 h, washed with 100% methanol, rehydrated with 50% methanol in PBSTw, followed by two PBSTw washes. Then, they were incubated in Proteinase K solution (2 µg/mL supplemented with 0.1% SDS in PBSTw) for 10 min without shaking and immediately post-fixed with 4% FA in PBSTw for 1 h.

For fluorescence in situ hybridization (FISH), after two PBSTw washes, and one wash in 50% pre-hybridization buffer, samples were incubated in pre-hybridization buffer (prehyb, 50% deionized formamide, 0.1% Tween-20, 5x SSC, 1% SDS in DEPC-treated water) at 56 °C in a hybridization oven for 2.5 h. Hybridization proceeded overnight at 56 °C in hybridization buffer (50% deionized formamide, 0.1% Tween-20, 5x SSC, 1% SDS, 0.1 mg/mL yeast RNA, 0.1 mg/mL salmon sperm DNA, 62.5 µg/mL of heparin, in DEPC-treated water; riboprobes diluted at 1:1000). Samples were sequentially washed in prehyb, 50% prehyb, 2x SSC supplemented with 0.1% Tween-20, and 0.2x SSC with 0.1% Tween-20 for 20-30 min each at 56 °C. Samples were brought back to room temperature during the last wash, and then washed twice with MABTx (11.6 g/L maleic acid, 8.8 g/L NaCl, pH 7.5, 0.1% TritonX). Blocking was performed in 10% horse serum in MABTx for 1 h, followed by incubation overnight at 4 °C in an antibody solution of anti-DIG-POD (Roche, cat. # 11207750910, 1:1000 in blocking buffer). We washed the samples five times in MABTx and five times in PBTx (1x PBS, 0.1% TritonX, 0.1% BSA), each for 20 min. We then incubated the samples in tyramide buffer (2 M NaCl + 0.01 M boric acid, pH 8.5) for 10 min, followed by a 10 min incubation in development buffer (20 µg/mL IPBA, 0.003% H₂O₂, 20 µg/mL TAMRA in tyramide buffer) in the dark. After washes in PBTx, samples were mounted in scale solution (30% glycerol, 0.1% TritonX, 4 M urea, 2 mg/mL sodium ascorbate, in PBS)[90] for imaging.

For hybridization chain reaction (HCR), samples were washed twice with PSBTw and then incubated with probe hybridization buffer (Molecular Instruments) for 30 min at 37 °C, before the hybridization buffer (probe hybridization buffer with 4 pM of the *Cl-runt* oligo pool) was added. The *Cl-runt* oligo pools were designed using the probe generator from the Ozoplat Lab[91] (Supplementary Data 4). Samples were incubated for 12 h, and washed four times with probe wash buffer (Molecular Instruments) at 37 °C and five times at room temperature with 5x SSCT (5x SSC, 0.1% Tween-20). Samples were then incubated in amplification buffer (Molecular Instruments) for 30 min and incubated in a hairpin solution (30 pM of each hairpin, heated to 95 °C and snap cooled in amplification buffer). Samples were washed multiple times with 5x SSCT and mounted in scale solution for imaging. FISH and HCR samples were imaged on a Zeiss LSM 800 confocal microscope. Confocal sections with optimal z spacing based on the Zen software were captured to image the entire thickness of the acoel and maximum intensity projections were generated.

### RNAi-mediated gene knockdown

Genes of interest were identified from the reference transcriptome and primers were designed for the predicted ORFs (Supplementary Data 4). Double stranded RNA (dsRNA) was synthesized using the pJC53.2 plasmid (Addgene Plasmid ID: 26536), as previously described[89], and resuspended in 50 µL MilliQ water. RNAi was performed through microinjection into the gut using a XenoWorks Digital Microinjector. Needles were pulled with a vertical micropipette puller (Sutter Instruments Model P-30, with settings heat: 750 and pull: 750). The dsRNA (~2–3 µg/µL) was diluted 1:1 with ASW and mixed with food coloring dyes for visual confirmation of successful delivery. *Cl-runt* RNAi injections were done every 2 days over a week (3 injections total). *Cl-egr* RNAi injections were done every 2–3 days for 3 weeks (9 injections). Animals were fed brine shrimp before every injection to induce the formation of the acoel gut cavity, and water was changed after every injection. As the negative control, acoels were injected with dsRNA of the *ccdB* and *camR* insert sequence in the pJC53.2 plasmid. Control RNAi was administered with the same frequency and time span as *Cl-runt* or *Cl-egr*, respectively, so that the differences between experimental and control groups can be attributed specifically to the knockdown, rather than the stress caused by the experimental procedure.

## Statistics and reproducibility

All experiments were repeated on different dates at least twice with more than three biological replicates. For RNAi experiments, regeneration was evaluated in both control and *Cl-runt* or *Cl-egr* dsRNA treated samples. If the control RNAi animals did not regenerate, the experiment was discarded. For PAM analysis, biological replicates were discarded if the fluorescence was not distinguishable from the background fluorescence. Investigators were not blinded to allocation during experiments and outcome assessments except for PAM measurements.

## Reporting summary

Further information on research design is available in the Nature Portfolio Reporting Summary linked to this article.

## Data availability

The reference transcriptomes generated in this study are provided in Supplementary Data 1, 2. The annotations generated in this study are provided in Supplementary Data 3. The RNA-seq datasets generated in this study have been deposited in the Gene Expression Omnibus (GEO) database under accession number GSE242841 [https://0-www-ncbi-nlm-nih-gov.brum.beds.ac.uk/geo/browse/?view=samples&series=242841&zsort=date&display=20&page=1], and through SRA under the project number PRJNA1015130. The normalized read count and log$_2$FoldChange values for all genes used in the figures are provided in Supplementary Data 5. Source data are provided with this paper.

## Code availability

The transcriptome assembly and annotation pipeline are available at www.github.com/xuesoso/acoel_reference_assembly.

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

## Acknowledgements

We thank Paul Bump for help with designing the oligo pools used for the *Cl-runt* HCR experiment, Jesse Gibson for help with the gDNA extraction protocol, Robin Rio Bigasin for help with animal husbandry, and all Wang group members for discussion. D.N.S. is a BioX Bowes Fellow, E.S.S. and Y.X. are Stanford Interdisciplinary Graduate Fellows. B.W. is a Beckman Young Investigator. A. Burlacot is supported by the Carnegie Institution for Science. The long-reads sequencing was performed at CZ Biohub. This work is supported by a NIH grant 1R35GM138061 to B.W. Some schematics were created with BioRender.com.

## Author contributions

Conceptualization: D.N.S., Y.X., J.S., and B.W.; Methodology: D.N.S., Y.X., A. Byrne, D.L., A. Burlacot, and J.S.; Investigation: D.N.S., Y.X., and E.S.S.; Formal analysis: D.N.S., Y.X., and E.S.S.; Validation: D.N.S.; Writing: D.N.S. and B.W. with feedback from Y.X., A. Burlacot, and J.S.; Funding acquisition: B.W.; Supervision: S.D., S.R.Q., A. Burlacot, J.S., and B.W.

## Competing interests

The authors declare no competing interests.
