## [Peer Review file · Nature Communications]

REVIEWERS' COMMENTS

Reviewer #1 (Remarks to the Author):

The authors have submitted an exceptionally strong revision of this paper. New data support the most essential points of the manuscript and add new depth and nuance throughout the work. I am supportive of publication of the manuscript as is and congratulate all of the authors on a really outstanding manuscript and body of work. It was very enjoyable to read and I look forward to reading and learning more as the authors build on the strong foundation they have created in this work.

Reviewer #2 (Remarks to the Author):

Thank you to the reviewers for addressing my concerns. There are several places where the manuscript was greatly improved. I also apologize for misreading the control RNAi figure. This section is now thoroughly discussed and will hopefully address that misunderstanding for future readers.

The live imaging experiments sound like they add significantly to the paper. However, I could not load the videos (unfortunately) on my computer. This might be my technical problem, but it is also important for the authors to check the videos before publication. The lack of directed migration is interesting and may provide insights into symbiosis maintenance in non-regeneration conditions.

Fig 2c: The significance threshold of $P < 0.1$ is not traditional. However, using a Tukey posthoc test is probably over-penalizing due to the nature of the time course. I think this is fine.

The additional RNAseq experiments suggest the process is much more dynamic than initially thought. I hope this gets explored in future research.

All other points were addressed satisfactorily by the reviewers.

Reviewer #2 (Remarks on code availability):

The data availability issues are resolved, and the methods are clear. I could not open the videos. This may be a problem on my side, but it is worth checking.

Reviewer #3 (Remarks to the Author):

The authors have thoughtfully addressed my concerns. The new data make the manuscript stronger and the findings therefore more significant for the understanding of the role of photosymbiosis in acoel regeneration. I recommend it for publication.

Response to reviewers

We thank the reviewers for their thoughtful and constructive reviews. In response to the suggestions, we have performed new experiments, provided new figures, and extensively revised the text. We have tried our best to address all of the referees' comments, and the manuscript has been improved in the process. The major changes in the text are highlighted in red.

Reviewer #1

Some species live in symbiotic pairings, coexisting with partners and coupling the responses of two genetically distinct populations of cells to promote the health of both species within a metaorganism. Cells of symbiotic partners must communicate with each other to promote coordinated and cooperative physiological responses. For example, after injury of a metaorganism consisting of distinct symbiotic species, a regenerative response is only likely to be successful if cells of both species act in harmonized manners to replace both cell types. Alternatively, regeneration of a metaorganism could proceed through regenerative action of one species and subsequent replacement of new cells of the second species from outside the organism. In this manuscript, the authors explore regeneration of a pair of partners that coexist within a metaorganism, the acoel *Convolutriloba longifissura* and algal cells of the genus *Tetraselmis*. The high regenerative capacity of acoels allowed the authors to investigate coordination of cellular behaviors between acoel and algal cells.

This paper sets out with a stated goal of understanding the post-injury response of algal and acoel cells in the regenerative metaorganism, which is an important and interesting goal. The authors complete transcriptomic analyses to explore this process, before focusing on changes in photosynthetic activity of the algae post-injury. Major strengths of the paper include the groundwork laid in exploring these coexistent species (e.g. transcriptomic and genomic data, methods development in this species) and the interesting biological properties of the system chosen. However, the authors conclude that changes in photosynthesis in the algal cells represent a "wound response" (title), despite the fact that changes in F_v/F_m and $Y(II)$ are widely reported as stress responses in other plant species. Thus, another reasonable interpretation of the data would be that the authors have demonstrated a wound-induced stress response in symbiotic algal cells.

We thank the reviewer for recognizing the strength of the work and appreciate their time and effort in reviewing this manuscript.

We now present two lines of evidence supporting the notion that the algal response is not a generic stress response. First, we demonstrate that light stress-induced algal responses are distinct from those induced by host injury (**Figure 4**). Although there is a small overlap, which may represent the generic stress responses elicited by both light stress and host injury, the primary responses are clearly divergent under these two conditions. This divergence is marked by the upregulation of many photosynthesis-related genes after injury that are instead downregulated under intense light. Intriguingly, the upregulation of some of these photosynthetic

genes has been described in stress-tolerant plants and tolerance primed plants. This observation, together with the fact that F_v/F_m stabilizes shortly after injury and only $Y(II)$ continues to decrease at 24 hpa (**Fig. 2c-e**), suggests that, contrary to stress responses aimed at containing photo-damage, the injury-induced transcriptional responses may serve to compensate for the loss in photosynthesis caused by host injury.

Second, following the reviewer's recommendation, we have performed additional RNA-seq experiments to evaluate the algal responses at later time points during host regeneration. Our findings, now described in the new **Figure 7** and **Supplementary Figure 8**, revealed that both the acoel and the algae exhibit a new wave of responses at 2 days post amputation (dpa), which differ significantly from their responses at early time points. Critically, similar to early responses, these late responses in the algae were fully diminished after *Cl-runt* RNAi, suggesting that the algal responses are coupled with the acoel's regeneration program. The reviewer's feedback has prompted us to clarify this nuanced but important point. We have now incorporated the new data (line **392-407**) and provided in-depth discussions of this point in the revised text (line **176-184, 263-269, 281-284, 307-313, 420-425, 430-438**).

Additionally, and more importantly, the authors show that photosynthesis and changes in photosynthetic activity are not required for regeneration, further weakening the argument that the photosynthetic changes observed are a key element of wound response in the metaorganism. Thus, while the authors have demonstrated something that occurs during regeneration in an intriguing pair of symbiotic organisms, it's not clear that the phenomenon chosen is fundamentally important in regeneration.

We agree with the reviewer that it may seem more intuitive for algal photosynthesis to be essential for host regeneration. However, in our opinion, the finding of photosynthesis not being necessary for regeneration is an equally important, if not more surprising, result. The significance of this result has also been acknowledged by Reviewer 2. In light of this, we have moved the data previously in **Supplementary Figure 2** to the main **Figure 2a**. We have also revised the text accordingly (line **144-149**).

In addition, we now discuss this observation explicitly in the text (line **501-510**), which reads:

“Finally, although algal photosynthesis responds to host injury, we found that algal photosynthesis is not required for acoel regeneration. This finding aligns with observations in other photosymbiotic animals²⁵ and may reflect the resilience of these systems, which likely build on several alternative energy sources. Besides glucose, photosynthetic algae can produce lipids for energy storage, sustaining their long-term growth and survival⁷⁸. *Tetraselmis* algae stand out for their lipid synthesis efficiency⁷⁹. It is conceivable that the holobiont taps into these lipid reserves for the energy required during regeneration. In addition, the acoel may directly consume algae, as recently observed for corals and their photosynthetic symbionts⁸⁰. Supporting this, we consistently observe the emergence of a feeding spot during regeneration^{28,29} (Supplementary Fig. 1).”

The fundamental questions we aimed to address here is whether the symbionts respond to host injury, and whether their responses are autonomous or regulated by the host. To our knowledge, this work is the first study addressing these important questions, establishing a causal link between the symbiont' molecular responses and a highly conserved gene regulatory program associated with the host regeneration. We have now modified the text to clarify the main motivation and contribution of this work (**line 31-50, 410-428**).

More mechanistic insight into how algal cells regenerate and/or how algal and acoel cells influence one another in the context of regeneration would be required for this work to merit publication in Nature Communications. I think this is possible and suggest several options for expanding the analysis below. Alternatively, the paper could be appropriate as written (or with minor changes) in a more specialized journal, should the authors so choose.

We have explored all the directions suggested by the reviewer, as detailed below. We have succeeded in making progress along several directions, which has substantially strengthened the paper, whereas others proved to be beyond our current technical capabilities, despite repeated attempts. These challenges highlight the need for developing new tools, which are important avenues for future research.

Major suggestions:

1) My primary critique is that the authors use F_v/F_m and $Y(II)$ to explore reduction in photosynthetic efficiency/activity of the algal cells. Broadly in the literature, these types of assays are very commonly used to indicate plant stress. The alternative hypothesis, that injury to the acoel stresses algal cells, is also supported by the data as presented. The alternative hypothesis is also supported by upregulation of many stress-related genes in the algal cells and by the observation that photosynthesis is not required for regeneration. Together, the evidence suggests that photosynthetic changes might indicate stress rather than play a function in regeneration. I would argue that other aspects of algal physiology and cell biology may be much more important for the context of acoel regeneration.

We have conducted new experiments that lend strong support to the notion that the algal responses are more than generic stress response. For the clear divergence between the responses induced by light stress and host injury, please see **Figure 4**. **Figure 7** presents new data showing that the algal cells exhibit a second wave of responses at later stages of host regeneration, which are distinctly different from their early responses. Please see our reply to the previous two questions for details.

To answer the reviewer's question regarding the implication of F_v/F_m and $Y(II)$ measurements, we have now modified the text (**line 172-184**) to read:

"We then evaluated the effects of amputation on photosynthesis in a matched timeframe (0 and 24 hpa). We added a 3 hpa time point as animal wound responses are expected to be already pronounced at this early time ^{12,32,33}, which may also induce changes in algal physiology. Both F_v/F_m and $Y(II)$ decreased concordantly at 3 hpa (Fig. 2c) but only $Y(II)$ continued to decrease at 24 hpa (Fig. 2e). This may be caused by further

interference with the electron transport chain's photoconversion capacity at this later time point without affecting the maximum fraction of functional open centers. This suggests that host injury primarily induces a decrease in the sink capacity for photosynthates downstream of PSII, thereby limiting electron flow and eventually decreasing F_v/F_m . It is noteworthy that, light stress reduces F_v/F_m typically through a different mechanism: excess light overexcites the photosynthetic machinery, saturating the photosynthetic electron transport chain and producing reactive oxygen species (ROS), which result in degradation of photosystems in a process called photo-damage^{40,41}."

We also agree with the reviewer that other aspects of algal physiology and cell biology may play a role in the host regeneration. Indeed, we have observed that many transporter proteins were differentially regulated after injury in a *Cl-runt*-dependent manner, suggesting that the exchange of nitrogen and carbon metabolites may lead to coordinated regeneration responses between species. This important point is now discussed extensively in the text (**line 449-492**).

Alternative ideas to consider for exploring the response of algal cells to injury would include:

a. Exploring how algal cells respond to wounding in ways that promote their own replacement or the successful regeneration of the metaorganism. Do algal cells undergo cell divisions? Do they label with pH3 or BrdU (Fig. 1g)? Can they be quantified per animal area at different time points to see if numbers change?

We agree that characterizing the proliferation dynamics of algae could offer valuable insights. However, our repeated attempts to stain algal cells with BrdU, EdU, or pH3 have been unsuccessful, which seems to be a common limitation in the area of algal research. Additionally, we have attempted to use flow cytometry to quantify DNA content in algal cells as a measure of their cell cycle progression, but the results appeared to be highly sensitive to dye concentration and staining time, leading to inconsistent and confounding results.

On a more positive note, we succeeded in quantifying the relative abundance of algal and acoel cells via cytometry. The ratio did not fluctuate significantly throughout the course of regeneration (new **Supplementary Figure 2**), suggesting that the algal density may be maintained during regeneration. However, it is also plausible that only a minor fraction of the cells undergo proliferation and turnover, which would not significantly alter the overall cell compositions.

b. Presenting data to show how algal cells enter the blastema site. Do they come from algae in the trunk or from outside the animal? Do they enter the blastema by migration, which might be seen through some type of live imaging?

Though descriptive, an experiment to explore this (perhaps using single regenerating animals in isolation with frequent water changes to see if restoration of algae is inhibited) would be helpful in understanding algal responses during regeneration.

We are grateful to the reviewer for this great suggestion. In response, we have developed a live imaging method capable of visualizing algal movements within the acoel tissue for short periods of time. In these experiments, acoels were anesthetized with magnesium chloride, which after

extended treatment resulted in acoel death, precluding long term imaging. Despite this limitation, the captured videos (now presented as **Supplementary Video 1, 2**, and illustrated in the new **Figure 1i**) are very informative, showing that the algae are quite motile. Notably, at 2 dpa, algal cells appeared to be mostly restricted from the blastema, yet by 3 dpa, they freely moved in and out of the newly regenerated tissue. We did not observe any instances of algal cells entering from the external environment. These results suggest that algal colonization of new tissues is not a result of directed migration but rather through random dispersal of algal cells from pre-existing tissues. We have updated the text accordingly to include this important result (**line 132-140**).

2) Alternatively, more mechanistic insight could be come from exploring why algae are required for acoel regeneration. What occurs when algae are absent? Can you infer what role algal cells play in metaorganism/acoeel cell regeneration (e.g., acoel cell division, polarity)? This might be known from the literature, and/or could be determined experimentally.

Currently, there is no evidence suggesting that the algae are required for acoel regeneration. In the attempt to address this question, we treated acoels with DCMU to eliminate algae. While this treatment led to animal death in a couple weeks (**Supplementary Figure 4**), the animals exhibited a range of regeneration capacity before death. However, interpreting this result is difficult, due to the potential non-specific effects caused by the deteriorating animal physiology following algal depletion. We now clarified this point in the text (**line 1264-1265**). As our study is the first investigating regeneration in this acoel species, this information is not available in the literature either.

3) One other potential limitation of the manuscript is that the choice of very early time points (e.g. 3 and 6 hpa – Fig. 2, 3A) may not be appropriate given that algal cells reenter the wound site much later, after 2 DAYS post amputation. Exploring algal transcriptomic changes at later time points might give a better view of algal response to injury that is relevant to algal participation in the regeneration process.

Following the reviewer's suggestion, we have now added RNA-seq experiments to analyze transcriptional changes in both acoel and algal cells at 1 and 2 dpa, now reported in **Figure 7**. Surprisingly, we observed minimal changes at 1 dpa compared to the baseline at 0 dpa for both acoel and algal genes, but hundreds of genes were differentially expressed at 2 dpa (**Figure 7a,b**). This suggests that, by 2 dpa, the system switched to a second wave of regenerative responses. Critically, genes in this second wave had little overlap with those responding at early time points (**Figure 7c**). For example, genes associated with photosynthesis, which were differentially expressed at 3 and 6 hpa, appeared to be specific to the early response, whereas a suite of algal transcription factors were upregulated only at the late time points (**Figure 7c,d**). Additionally, we identified algal transporters specifically responding at later stages of regeneration, which could suggest that the exchange mechanisms between acoel and algae shift as regeneration progresses (**Figure 7d, Supplementary Figure 7d,e, and Supplementary Figure 8e**). Finally, we found that this second wave of responses in both acoel and algae was blocked by *Cl-runt* RNAi, suggesting that the gene regulatory program centered around *Cl-runt*

is necessary for the coordination of acoel and algal responses at both the early and late phases of regeneration. We have modified the text to include this new result (**line 392-407**).

4) Finally, the only gene perturbation shown to impact both cell types was *runt-1*(RNAi) which was also reported to impact regeneration in *Hofstenia*. Further exploration of why and how this RNAi impacts algal cells would be an interesting and third possible path forward in providing mechanistic insight. It seems important to show that *runt-1*(RNAi) alters some aspect of algal biology that impacts metaorganism regeneration. First, the impacts on Fv/Fm and Y(II) are either small or not significant at 24 h, so this might not be the main way that algal cells are impacted. Second, the authors show transcriptional changes at 3 h, but do not present functional data (including Fv/Fm and Y(II)) at this time point.. Does algal number/density change? Algal division? The authors also emphasize changes in transporter gene expression, but metabolomic analysis of acoel or algal cells might help to elucidate the possible importance of these changes. These are several options, which collectively are likely beyond the scope of this paper, but seeking out one among these options could be a path forward if the authors choose to resubmit to Nature Communications.

While the changes in Y(II) caused by *Cl-runt* RNAi are small, they are statistically significant and highly reproducible between biological and technical replicates. These changes could potentially be more pronounced under specific environmental conditions and additional stressors that acoels may encounter in their natural habitats. It is plausible as well that *Cl-runt* influences other aspects of algal biology, however, the lack of tools for assessing algal physiology and proliferation presents a challenge in investigating these aspects further.

In determining the time points for measuring photosynthetic activity, we chose 24 hpa because, at this time, F_v/F_m and Y(II) showed more pronounced responses compared to 3 hpa (**Figure 2**). A synchronization of transcriptional and functional responses is not expected due to the time needed to translate changes in mRNA abundance into observable functional outcomes. Indeed, our new data (**Figure 7**) suggest that there was a small transcriptional response at 24 hpa, even though the changes in photosynthesis were most evident. This important point is now clarified in the text (**line 606-611**).

Following the reviewer's suggestion, we measured algal density by quantifying the ratio between algal and animal cells by flow cytometry. We found no difference between control and *Cl-runt* RNAi samples, suggesting that *Cl-runt* RNAi did not change the algal content at the population level.

Finally, we have also tried to apply metabolomic analysis to this system, but the mass spectrometry signal has been so far dominated by an unknown small molecule metabolite. We have recently initiated a collaboration with experts in the metabolomic analysis field and hope to follow up on this in a future study.

Minor suggestions:

1) A nomenclature to help the reader distinguish between Convolutriloba and Tetraselmis genes would be helpful (e.g. Cl-runt or similar). This will help the reader understand which genes being discussed or investigated belong to which organism.

Done.

2) One minor critique of this manuscript is that it is hard to know what was already known about Convolutriloba regeneration, interdependence of Convolutriloba and Tetraselmis, etc. A stronger grounding of this paper in the context of published literature would help the reader to appreciate necessary background and to better place this work into that context.

This study is the first examining regeneration in this species at the molecular and functional levels, and stands as one of the first to investigate molecular underpinnings of animal-algal symbiotic relationships during regeneration in any photosymbiotic holobiont. We have now made this point clear in the text (**line 63-65, 75-76, 410-411**) – thanks for pointing this out.

Reviewer #2

Summary:

The current manuscript takes an interesting approach to understand the relationship between algal symbionts and regeneration. The authors combine well-established methods in algal physiology and acoel worm regeneration. The relationship between microbes in animal development and biology is of considerable interest, so I believe this paper would be well-received.

In particular, the manuscript presents data supporting the hypothesis that there are dynamic changes in algal physiology and gene expression upon injury to the acoel host. The authors identified that *runt*, a well-studied transcription factor required for whole-body regeneration in an acoel and planaria, may alter algal transcription in response to injury. Overall, the manuscript is well-written, with clear figures. I have important questions about the result that *runt* RNAi impacts algal photosynthesis and transcription. It appears possible that the downregulation of photosynthesis may be a general stress response (i.e., from injection) and not due to the gene knockdown. Additionally, the authors overstate the results in some areas, as highlighted below. However, I anticipate the authors could make these adjustments easily.

We thank the reviewer for the positive comments, and appreciate their time and efforts in reviewing this manuscript.

Regarding to reviewer's concerns about the potential stress from injection influencing changes observed with *runt* RNAi, we would like to clarify that control RNAi animals were also injected with dsRNA of a non-acoele sequence. Therefore, the differences between *runt* and control RNAi animals can be attributed specifically to the knockdown, rather than the stress caused by the experimental procedure. We have clarified this important point in the methods section (**line 804-808**).

Main points:

- The lack of requirement of algal photosynthesis on regeneration appears to be a key piece of data that is subjected to the supplement and not in the abstract. This has important implications for how to interpret the transcriptional changes. I would recommend moving Figure S2 to Figure 1 (maybe to replace 1I?). Is there quantification or replication of the experiments in Figure S2?

Thank you for this great suggestion. We have moved this result from the previous **Supplementary Figure 2** to the main **Figure 2** and clarified the importance of this result in the abstract and the main text (**line 24, 142-149, 501-510**). We have quantified the number of animals that regenerated normally: under light, 19 out of 21 animals regenerated normally, while, in the dark, 33 out of 36 regenerated normally. This information is now reported in **Figure 2a**.

- The manuscript would be improved if the authors generated a more careful description and quantification of algae during regeneration to explain the distribution and dynamism (if any) of algal cells. Is it possible to visualize if algal number or dynamics change during regeneration with the data already collected?

We developed a live imaging method capable of visualizing algal movements within acoel tissues. In these experiments, acoels were anesthetized with magnesium chloride, which after extended treatment resulted in acoel death, precluding long term imaging. Despite this limitation, the captured videos (now presented as **Supplementary Video 1, 2**, and illustrated in the new **Figure 1i**) are very informative, showing that the algae are quite motile. Notably, at 2 dpa, algal cells appeared to be mostly restricted from the blastema, yet by 3 dpa, they freely moved in and out of the newly regenerated tissue. These results suggest that algal colonization of new tissues is not a result of directed migration but rather through the random dispersal of algal cells from pre-existing tissues. We have updated the text accordingly to include this important new result (**line 132-139**).

We have also measured algal density by quantifying the relative ratio of algal cells to acoel cells using flow cytometry. We found no difference throughout the course of regeneration (**Supplementary Figure 2**), suggesting no major change in algal content at the population level. However, it is also plausible that only a minor fraction of the cells undergo turnover, which would not significantly alter the overall abundance.

- The significantly different relationships in Figure 1J are not clear. Are time points 3 and 24 each significantly different from time point 0, or are the 3 and 24 hpa conditions only significant when pooled?

We apologize for the confusion and have made this clear in the figure and modified the caption to read:

“All individual treatments are significantly different compared to the ‘0 hpa’ data, whereas there is no significance between any other conditions (one-way ANOVA and a Tukey post hoc test). “

- Are PAM measurements consistent regardless of the number of algae present in a tissue fragment or its size? Can you comment in the results on the location of the algae that you are measuring in the PAM and RNAseq experiments. It appears, both for the Fv/Fm and RNAseq, that the impacts of amputation are systemic. Is that true?

In PAM measurements, both F_v/F_m and $Y(II)$ derive from the normalized ratio between variable and maximum fluorescence. This removes the dependence of these metrics on the number of algae, as long as the sample size is sufficient to generate consistent fluorescence signal. When optimizing the PAM measurement on our acoels, we determined the number of animals needed to achieve consistent signal. We found that 20 tail fragments per measurement gave consistent

results and therefore use this number for all our experiments. We have now added the following text to the methods (**line 582-584**):

“We used 20 tail fragments for each replicate in all experiments. This number was determined to give sufficient fluorescence signal allowing for reliable quantification of F_v/F_m and $Y(II)$. “

Both PAM measurements and RNAseq experiments were performed on groups of tails, which lack spatial resolution. We now clarified in the text that these are “bulk measurements” or measurements “at a population level”. Measuring these responses in a spatially resolved manner is indeed an important avenue for future research.

- Is there a no-amputation and no-light-stress control time course for Figure 1J? It is common for photosynthetic parameters to change when making subtle changes to husbandry and/or environmental conditions.

We found that tails yielded most reproducible results compared to heads or intact animals. The “0 hpa, no light stress” control is the best control condition since algae in the anterior may have different photosynthetic activity. This control condition also includes all the handling steps experienced by other samples, ensuring a uniform baseline for comparison. Amputation took less than five minutes and was done under green light after dark incubation to minimize perturbations to the system as much as possible. These experiments were repeated multiple times on different dates to ensure that the results are not due to any unaddressed variables. We have now clarified these important technical details in the methods (**line 597-604**).

- It is not clear which datasets were compared to find shared DEGs between *C. long.* and *H. mia*.

This is now clarified in the figure caption of **Figure 5**:

“Comparison of \log_2 FoldChange (FC) in expression of orthologous genes at 3 vs 0 hpa between *C. longifissura* and *H. miamia*. One-to-one orthologs were identified using reciprocal BLAST. Differentially expressed genes at 3 hpa for *H. miamia* were obtained from Gehrke, *et al.*, 2019³².”

- I have several questions regarding the expression of *runt* in *C. long* and the RNAi phenotypes. First, is *runt* expressed in intact and asexually dividing *C. long*? Second, is there wound closure in the *runt* RNAi animals? There does not seem to be a blastema. Third, how many *runt* orthologs are there in *C. long*?

As shown in **Figure 5b** and **Supplementary Figure 6c**, *runt* expression was minimally expressed in animals at 0 hpa. We did not examine its expression in asexually dividing animals, as we could not predict when fission might initiate.

The *runt* RNAi animals did close the wound but didn't form blastema, we have now modified the text (**line 334-338**) to read:

“Although all *Cl-runt* RNAi animals closed the wound, none of the tail fragments formed a clear blastema or regenerated a head, evidenced by the absence of the neural ganglion at 4 dpa along with other head structures (**Fig. 5c,d**), whereas only ~15% of *Cl-runt* RNAi treated anterior fragments failed to regenerate a tail (**Supplementary Fig. 5d**). “

There is only a single *runt* homologue in *C. longifissura*, we have now clarified this in the text (**line 326-328**).

- Are the relative fold-changes between 0 -3 hpa different between the control RNAi and the *runt* RNAi? The z-scored values make it hard to tell. It looks like the *runt* RNAi 0 hpa might just start at a lower level?

Thanks for raising this important point. Only three genes (*psbs-4*, *bst-2*, *pgr5*) were differentially expressed at 0 hpa between the control and *runt* RNAi animals, suggesting that *runt* influences the photosynthesis pathway mostly during regeneration. We have revised the text to highlight this point (**line 362-364**). Furthermore, we have now included **Supplementary Data 3**, which reports all the normalized expression and log₂ FoldChange values of the genes used in all heatmaps throughout the figures.

- Finally, it appears that injection causes a decrease in F_v/F_m compared to the controls. There does seem to be an impact with the *runt* RNAi in Y(II), however the difference is subtle. Could any stress mechanical or light just impact algal photosynthesis? The direct relationship between *runt* knockdown and algal physiology could be stronger.

The RNAi protocol indeed induced a decrease in F_v/F_m, as photosynthesis is sensitive to many environmental conditions. Therefore, in our experiments, we made sure that the control RNAi animals also received injections with the same schedule as the *runt* RNAi group, and these two groups of animals were maintained in the same condition and always handled in parallel, to rule out the influences of other variables. While the changes in Y(II) caused by *runt* RNAi are small, they are statistically significant and highly reproducible between biological and technical replicates. We have now included more technical details of the RNAi experiments in the methods section (**line 804-808**).

To further evaluate the relationship between *runt* knockdown and algal physiology, we have performed additional RNA-seq experiments to measure gene expression changes at later time points (1 and 2 dpa) during regeneration between control and *runt* RNAi. Surprisingly, we observed minimal changes at 1 dpa compared to the baseline at 0 dpa for both acoel and algal genes, but hundreds of genes were differentially expressed at 2 dpa (**Figure 7a,b**). This suggests that, by 2 dpa, the system switched to a second wave of regenerative responses. Genes in this second wave had little overlap with those responding at early time points (**Figure**

7c). For example, genes associated with photosynthesis, which were differentially expressed at 3 and 6 hpa, appeared to be specific to the early response, whereas a suite of algal transcription factors were upregulated only at the late time points (**Figure 7c,d**). Additionally, we identified algal transporters specifically responding at later time points, which could suggest that the exchange mechanisms between acoel and algae may shift as regeneration progresses. Critically, we found that this second wave of responses in both acoel and algae was blocked by *runt* RNAi, suggesting that the gene network centered around *runt* is necessary for the coordination of the acoel and algae responses throughout regeneration. We have modified the text to include this new result (**line 392-407**), and discussed this important point extensively (**line 420-428, 462-492**).

Minor points:

- It seems that anterior regeneration must require algal repopulation as new tissue is formed. What are the key differences between anterior and posterior? Is it just that there is a blastema that is initially cleared of algae?

As the reviewer pointed out, there are key differences between anterior and posterior regeneration. During posterior regeneration, (1) blastema did not form and there was no major accumulation of dividing cells (**Fig. 1f, g**), (2) the gross morphology of the tail recovered rapidly by 2 dpa (**Fig. 1f**), and (3) *runt* expression was not induced at the wound site (**Supplementary Fig. 6c**) and it is not required for posterior regeneration (**Supplementary Fig. 6d**). The features of posterior regeneration align with morphallaxis, which is characterized by tissue remodeling without substantial cell proliferation. The differences between anterior and posterior regeneration are described in the text (**line 123-130, 330-338**).

- In the DCMU treatments and regeneration experiments, are animals eating or being fed? Did the authors confirm the absence of algae after DCMU treatment through fluorescence imaging?

In our regeneration experiments, the acoels were not fed. For the long-term DCMU treatments the animals were fed every other day until amputation. We confirmed the absence of algae after DCMU treatment through fluorescence imaging. The following text has been added to the methods (**line 567-572**):

“During the treatment, the animals were kept unfed. Effects of the DCMU treatment were confirmed with a Dual-PAM system, showing blockage of PSII (Supplementary Fig. 3c). For long-term DCMU treatments, animals were treated with DCMU twice, three days apart, and fed every other day. Algae were completely removed from the acoel two weeks after the two treatments, validated through imaging using the red fluorescence of the algae at 633 nm.”

- The high-light treatment on Fv/Fm is rather subtle. Does this get stronger if you use a higher light? Why was this light level chosen? I do not think a new experiment would add much to the

paper. I am just curious. I do think it is important to put all the models of the light sources used in the method.

The algae appear to have acclimatized to the light intensity at which they have been maintained, $50 \mu\text{mol m}^{-2} \text{s}^{-1}$. Above this light intensity, we observed a drastic drop in Y(II) (**Supplementary Fig. 3d**). For the light stress experiment, we tripled the light intensity to $150 \mu\text{mol m}^{-2} \text{s}^{-1}$, which is the maximum light intensity before the system got completely saturated, making it impossible to differentiate the changes in fluorescence and the background noise (as observed in **Supplementary Figure 3**). These conditions are now detailed in the text (**line 162-165**).

- Briefly defining “morphallactic” and “epimorphosis” might help the reader.

We now define these two terms in the text (**line 119-121, 128-130**).

- It is common to use z-scored heatmaps for RNAseq to maximize the visual differences in the expression of genes. However, this masks the actual fold-change values, which may be subtle. To help the readers assess the robustness of the RNAseq data, I suggest that the authors report all expression and fold-change comparisons for all of the genes they highlight in the main figures in the supplement.

We have added the normalized read count and \log_2 FoldChange values for all genes used in figures in **Supplementary Data 3**. The \log_2 FoldChanges are typically in the range of 1 to 2.5.

- The GitHub link for the assemblies was not active using Chrome or Safari browsers.

We apologize, the link should be active now.

- Figure 3C, D: Is there a legend for these heat maps? It might be the same as Figure 3B, but I am not sure.

Yes, they are the same. For clarity, we have added the legend to all panels.

- Figure 3e: The biological replicates show some consistency, but there are some replicates that look very different from the other replicates (e.g., 0 LL first row and the 24 HL first row). If visualized as fold-change, is there interesting biology here?

We pooled 5 animals for each sample to reduce potential noise caused by individual variation, but some degree of variation is still possible. We did look at the data but didn't observe consistent trends.

- Supplementary Figure 6A: What is the x-axis on the heat maps?

We thank the reviewer for catching this. These are individual samples at different time points. We have fixed the figure.

- Supplementary Figure 6B: It is difficult to make out morphology and true gene expression versus background. How many fragments were visualized in these experiments?

We have now changed the images to better show the expression of *egr*. We have also added the number of fragments examined in these experiments.

- Supplementary Figure 6C: How many fragments were visualized in these experiments?

We now added this information to the panels, 13 animals were examined at 0 hpa, and 14 animals at 3 hpa.

Reviewer #3

The role of symbiosis in metazoan regeneration is widely understudied so I was really excited when I was asked to review the paper. This is a beautifully written paper worth it of publication in Nature Communications. The study uncovers the role of the runt transcription factor (TF) in regeneration in a photosynthetic acoel, a marine invertebrate worm of contentious phylogenetic placement but important in the broader context of the evolution of regeneration in Bilateria. The photosymbiont is found extracellularly in between host cells. The runt TF seems to have a critical role in regulating not only the host regeneration program but possibly by still unknown regulatory mechanisms of the in hospite algal life cycle, as the host tissue regenerates. Through a detailed transcriptome analysis, several candidate downstream genes are identified in both partners in wounding and regeneration time series experiments under different light-intensity treatments. The methods and interpretation of the different data types (e.g., photobiology, etc) are explained in thoughtful detail, something I appreciated very much despite being familiar with most of the approaches used. That makes the manuscript easy to read and follow the conclusions.

We thank the reviewer for their excitement and careful reading of the manuscript.

Title: I have a minor quibble with the title. I believe that “metaorganism” is a large word that has been surrounded by quite a bit of debate. The authors put it in the title but then largely ignore it in the manuscript only mentioning it twice. Minimally you should define it precisely because of the different perspectives on what a metaorganism is. I believe that expanding on that would broaden the readership of the manuscript. I would encourage you to delve in the holobiont/metaorganism literature a little bit, and examine the papers by Seth Bordenstein and Kevin Theis (e.g., <https://journals.plos.org/plosbiology/article?id=10.1371/journal.pbio.1002226>), as well as Scott Gilbert’s on the subject. Bordenstein and Theis have worked extensively in defining the boundaries of the holobiont concept (e.g., <https://journals.asm.org/doi/full/10.1128/mSystems.00028-16>), and Gilbert tends to think about holobionts in the context of development (and by association, of regeneration) (e.g., <https://www.journals.uchicago.edu/doi/full/10.1086/668166>).

We thank the reviewer for this great suggestion. Following the suggestion, we have changed the title, and extensively rewritten the introduction and discussion to highlight this important concept (line 31-50, 410-428, 512-524). We have also cited the papers mentioned here in the references.

Abstract: Given that you use the term in the title, I believe you could frame your research and findings in a broader eco-evo-devo context. One sentence at the end of the abstract could possibly achieve that.

Thanks for this suggestion, we have now added the following sentence to our abstract:

“Our results suggest that during regeneration, the holobiont functions as an integrated unit of biological organization by coordinating gene regulatory networks across species

through the *runt*-dependent animal regeneration program.”

Introduction: Again in this section, mentioning 1) how common photosymbiosis is in metazoans and give a few examples; 2) how widespread regeneration is across animal phyla and better studied in some taxa vs others; and finally 3) that many photosymbiotic species regenerate pointing out how understudied that is and hence this magnificent paper. This citation may be helpful for that (<https://doi.org/10.1007/s13199-023-00920-0>). Also a couple of sentences in the life history of your acoele would be useful to the non-expert.

Thanks for these great suggestions. We have now extensively rewritten the introduction accordingly (**line 31-76**).

Results: I think you do a very good job in presenting the data and taking the reader to why you zoom into *runt* and its role in regeneration. I would suggest you define epimorphosis and morphallaxis.

Thanks, we now define these two terms in the text (**line 119-121, 128-130**).

Discussion: I suggest you consider a final paragraph where you take the reader back out to thinking about animal regeneration and the role of associated microbes, and in this case photosymbionts, in this developmental process. There are many animal hosts that regenerate and they are all holobionts that we can now dissect with the very same tools you use in this paper!

Thanks, we have now added the final paragraph to read (**line 512-524**):

“The interplay between hosts and endosymbionts during animal development has become increasingly recognized for its significance in the evolution and function of multicellular life forms^{7, 81}. Our approach offers a broadly applicable strategy to study the regulatory mechanisms by which animal hosts influence symbionts’ responses during critical developmental processes such as regeneration. By blocking host development and analyzing the symbionts’ response, we can determine whether symbionts contribute to the host’s developmental processes, and how they adjust their own physiology to ensure their survival in the changing environment within the host tissue. Many photosymbiotic animals, including *C. longifissura*, develop in the absence of their symbionts and gain their symbionts in later stages of their life cycle⁸². Evaluating the differences between embryonic development and regeneration, as well as comparing regeneration between symbiotic and non-symbiotic species, could help to understand how gene networks are integrated across symbiotic partners in holobionts and how these integrations may have emerged over evolution.”

Methods: Why did you feed the acoeles every three days and no other feeding cycle? How did you choose the light intensity for your experiments? Was PAM performed at the same time of the light:dark cycle during the different time points? Were the acoeles in the dark to measure respiration?

This feeding cycle was chosen to be consistent across experiments, but can be adjusted through a broad range. We did not explore the effects of feeding frequency, which didn't appear to affect regeneration.

The algae appear to have acclimatized to the light intensity at which they have been maintained, $50 \mu\text{mol m}^{-2} \text{s}^{-1}$. For the light stress experiment, we tripled the light intensity to $150 \mu\text{mol m}^{-2} \text{s}^{-1}$, which is the maximum light intensity before the system got completely saturated, making it impossible to differentiate the changes in fluorescence and the background noise (as observed in **Supplementary Figure 3d**). These conditions are now detailed in the text (**line 162-165**).

PAM measurements were performed at the same time during the day. Acoels were incubated in the dark prior to measuring F_v/F_m to ensure that all functional photosystem II are in a state ready to absorb light, which is required for calculating their maximum quantum yield. We have now added the following to our methods section (**line 597-604**):

“Freshly amputated tails were used as controls, as they undergo the same handling procedures as other experimental groups, ensuring a consistent baseline for comparison. This approach also ruled out potential differences in algal photosynthesis between anterior and posterior tissues. Amputations were completed quickly in under five minutes, and performed under green light following dark incubation to minimize perturbations to the system. To enhance reproducibility of our measurements, these experiments were repeated multiple times on different dates, but consistently at the same time of day, to ensure that the results were not influenced by unaddressed variables.”

Spell out WISH, FISH and HCR.

These technique abbreviations have been spelled out at their first usage in the manuscript.

Supplementary Figure 6 What are the 8 timepoints in 6a?

Thank you for catching this, each column is a sample in the RNA-seq experiment. We have modified the figure to fix this.

REVIEWER COMMENTS

Reviewer #1 (Remarks to the Author):

Some species live in symbiotic pairings, coexisting with partners and coupling the responses of two genetically distinct populations of cells to promote the health of both species within a metaorganism. Cells of symbiotic partners must communicate with each other to promote coordinated and cooperative physiological responses. For example, after injury of a metaorganism consisting of distinct symbiotic species, a regenerative response is only likely to be successful if cells of both species act in harmonized manners to replace both cell types. Alternatively, regeneration of a metaorganism could proceed through regenerative action of one species and subsequent replacement of new cells of the second species from outside the organism. In this manuscript, the authors explore regeneration of a pair of partners that coexist within a metaorganism, the acoel *Convolutriloba longifissura* and algal cells of the genus *Tetraselmis*. The high regenerative capacity of acoels allowed the authors to investigate coordination of cellular behaviors between acoel and algal cells.

This paper sets out with a stated goal of understanding the post-injury response of algal and acoel cells in the regenerative metaorganism, which is an important and interesting goal. The authors complete transcriptomic analyses to explore this process, before focusing on changes in photosynthetic activity of the algae post-injury. Major strengths of the paper include the groundwork laid in exploring these coexistent species (e.g. transcriptomic and genomic data, methods development in this species) and the interesting biological properties of the system chosen. However, the authors conclude that changes in photosynthesis in the algal cells represent a “wound response” (title), despite the fact that changes in Fv/Fm and Y(II) are widely reported as stress responses in other plant species. Thus, another reasonable interpretation of the data would be that the authors have demonstrated a wound-induced stress response in symbiotic algal cells.

Additionally, and more importantly, the authors show that photosynthesis and changes in photosynthetic activity are not required for regeneration, further weakening the argument that the photosynthetic changes observed are a key element of wound response in the metaorganism. Thus, while the authors have demonstrated something that occurs during regeneration in an intriguing pair of symbiotic organisms, it's not clear that the phenomenon chosen is fundamentally important in regeneration.

More mechanistic insight into how algal cells regenerate and/or how algal and acoel cells influence one another in the context of regeneration would be required for this work to merit publication in *Nature Communications*. I think this is possible and suggest several options for expanding the analysis below. Alternatively, the paper could be appropriate as written (or with minor changes) in a more specialized journal, should the authors so choose.

Major suggestions:

1) My primary critique is that the authors use Fv/Fm and Y(II) to explore reduction in photosynthetic efficiency/activity of the algal cells. Broadly in the literature, these types of assays are very commonly used to indicate plant stress. The alternative hypothesis, that injury to the acoel stresses algal cells, is also supported by the data as presented. The alternative hypothesis is also supported by upregulation of many stress-related genes in the algal cells and by the observation that photosynthesis is not required for regeneration. Together, the evidence suggests that photosynthetic changes might indicate stress rather than play a function in regeneration. I would argue that other aspects of algal physiology and cell biology may be much more important for the context of acoel regeneration. Alternative ideas to consider for exploring the response of algal cells to injury would include:

a. Exploring how algal cells respond to wounding in ways that promote their own replacement or the successful regeneration of the metaorganism. Do algal cells undergo cell divisions? Do they label with pH3 or BrdU (Fig. 1g)? Can they be quantified per animal area at different time points to see if numbers change?

b. Presenting data to show how algal cells enter the blastema site. Do they come from algae in the trunk or from outside the animal? ? Do they enter the blastema by migration, which might be seen through some type of live imaging?

Though descriptive, an experiment to explore this (perhaps using single regenerating animals in isolation with frequent water changes to see if restoration of algae is inhibited) would be helpful in understanding algal responses during regeneration.

2) Alternatively, more mechanistic insight could be come from exploring why algae are required for acoel regeneration. What occurs when algae are absent? Can you infer what role algal cells play in metaorganism/acoeel cell regeneration (e.g., acoel cell division, polarity)? This might be known from the literature, and/or could be determined experimentally.

3) One other potential limitation of the manuscript is that the choice of very early time points (e.g. 3 and 6 hpa – Fig. 2, 3A) may not be appropriate given that algal cells reenter the wound site much later, after 2 DAYS post amputation. Exploring algal transcriptomic changes at later time points might give a better view of algal response to injury that is relevant to algal participation in the regeneration process.

4) Finally, the only gene perturbation shown to impact both cell types was runt-1(RNAi) which was also reported to impact regeneration in Hofstenia. Further exploration of why and how this RNAi impacts algal cells would be an interesting and third possible path forward in providing mechanistic insight. It seems important to show that runt-1(RNAi) alters some aspect of algal biology that impacts metaorganism regeneration. First, the impacts on Fv/Fm and Y(II) are either small or not significant at 24 h, so this might not be the main way that algal cells are impacted. Second, the authors show transcriptional changes at 3 h, but do not present functional data (including Fv/Fm and Y(II)) at this time point.. Does algal number/density change? Algal division? The authors also emphasize changes in transporter gene expression, but metabolomic analysis of acoel or algal cells might help to elucidate the possible importance of these changes. These are several options, which collectively are likely beyond the

scope of this paper, but seeking out one among these options could be a path forward if the authors choose to resubmit to Nature Communications.

Minor suggestions:

1) A nomenclature to help the reader distinguish between *Convolutriloba* and *Tetraselmis* genes would be helpful (e.g. Cl-runt or similar). This will help the reader understand which genes being discussed or investigated belong to which organism.

2) One minor critique of this manuscript is that it is hard to know what was already known about *Convolutriloba* regeneration, interdependence of *Convolutriloba* and *Tetraselmis*, etc. A stronger grounding of this paper in the context of published literature would help the reader to appreciate necessary background and to better place this work into that context.

Reviewer #2 (Remarks to the Author):

Summary:

The current manuscript takes an interesting approach to understand the relationship between algal symbionts and regeneration. The authors combine well-established methods in algal physiology and acoel worm regeneration. The relationship between microbes in animal development and biology is of considerable interest, so I believe this paper would be well-received.

In particular, the manuscript presents data supporting the hypothesis that there are dynamic changes in algal physiology and gene expression upon injury to the acoel host. The authors identified that runt, a well-studied transcription factor required for whole-body regeneration in an acoel and planaria, may alter algal transcription in response to injury. Overall, the manuscript is well-written, with clear figures. I have important questions about the result that runt RNAi impacts algal photosynthesis and transcription. It appears possible that the downregulation of photosynthesis may be a general stress response (i.e., from injection) and not due to the gene knockdown. Additionally, the authors overstate the results in some areas, as highlighted below. However, I anticipate the authors could make these adjustments easily.

Main points:

- The lack of requirement of algal photosynthesis on regeneration appears to be a key piece of data that is subjected to the supplement and not in the abstract. This has important implications for how to interpret the transcriptional changes. I would recommend moving Figure S2 to Figure 1 (maybe to replace 1I?). Is there quantification or replication of the experiments in Figure S2?
- The manuscript would be improved if the authors generated a more careful description and quantification of algae during regeneration to explain the distribution and dynamism (if any) of algal cells. Is it possible to visualize if algal number or dynamics change during regeneration with the data already collected?
- The significantly different relationships in Figure 1J are not clear. Are time points 3 and 24 each significantly different from time point 0, or are the 3 and 24 hpa conditions only significant when pooled?
- Are PAM measurements consistent regardless of the number of algae present in a tissue fragment or its size? Can you comment in the results on the location of the algae that you are measuring in the PAM and RNAseq experiments. It appears, both for the Fv/Fm and RNAseq, that the impacts of amputation are systemic. Is that true?
- Is there a no-amputation and no-light-stress control time course for Figure 1J? It is common for photosynthetic parameters to change when making subtle changes to husbandry and/or environmental conditions.
- It is not clear which datasets were compared to find shared DEGs between *C. long.* and *H. mia*.
- I have several questions regarding the expression of *runt* in *C. long.* and the RNAi phenotypes. First, is *runt* expressed in intact and asexually dividing *C. long.*? Second, is there wound closure in the *runt* RNAi animals? There does not seem to be a blastema. Third, how many *runt* orthologs are there in *C. long.*?
- Are the relative fold-changes between 0 -3 hpa different between the control RNAi and the *runt* RNAi? The z-scored values make it hard to tell. It looks like the *runt* RNAi 0 hpa might just start at a lower level?
- Finally, it appears that injection causes a decrease in Fv/Fm compared to the controls. There does seem to be an impact with the *runt* RNAi in Y(II), however the difference is subtle. Could any stress mechanical or light just impact algal photosynthesis? The direct relationship between *runt* knockdown and algal physiology could be stronger.

Minor points:

- It seems that anterior regeneration must require algal repopulation as new tissue is formed. What are the key differences between anterior and posterior? Is it just that there is a blastema that is initially cleared of algae?
- In the DCMU treatments and regeneration experiments, are animals eating or being fed? Did the authors confirm the absence of algae after DCMU treatment through fluorescence imaging?

- The high-light treatment on Fv/Fm is rather subtle. Does this get stronger if you use a higher light? Why was this light level chosen? I do not think a new experiment would add much to the paper. I am just curious. I do think it is important to put all the models of the light sources used in the method.
- Briefly defining “morphallactic” and “epimorphosis” might help the reader.
- It is common to use z-scored heatmaps for RNAseq to maximize the visual differences in the expression of genes. However, this masks the actual fold-change values, which may be subtle. To help the readers assess the robustness of the RNAseq data, I suggest that the authors report all expression and fold-change comparisons for all of the genes they highlight in the main figures in the supplement.
- The GitHub link for the assemblies was not active using Chrome or Safari browsers.
- Figure 3C, D: Is there a legend for these heat maps? It might be the same as Figure 3B, but I am not sure.
- Figure 3e: The biological replicates show some consistency, but there are some replicates that look very different from the other replicates (e.g., 0 LL first row and the 24 HL first row). If visualized as fold-change, is there interesting biology here?
- Supplementary Figure 6A: What is the x-axis on the heat maps?
- Supplementary Figure 6B: It is difficult to make out morphology and true gene expression versus background. How many fragments were visualized in these experiments?
- Supplementary Figure 6C: How many fragments were visualized in these experiments?

Reviewer #3 (Remarks to the Author):

The role of symbiosis in metazoan regeneration is widely understudied so I was really excited when I was asked to review the paper. This is a beautifully written paper worth it of publication in Nature Communications. The study uncovers the role of the runt transcription factor (TF) in regeneration in a photosynthetic acoel, a marine invertebrate worm of contentious phylogenetic placement but important in the broader context of the evolution of regeneration in Bilateria. The photosymbiont is found extracellularly in between host cells. The runt TF seems to have a critical role in regulating not only the host regeneration program but possibly by still unknown regulatory mechanisms of the in hospite algal life cycle, as the host tissue regenerates. Through a detailed transcriptome analysis, several candidate downstream genes are identified in both partners in wounding and regeneration time series experiments under different light-intensity treatments. The methods and interpretation of the different data types (e.g., photobiology, etc) are explained in thoughtful detail, something I appreciated very much despite being familiar with most of the approaches used. That makes the manuscript easy to read and follow the conclusions.

Title: I have a minor quibble with the title. I believe that “metaorganism” is a large word that has been surrounded by quite a bit of debate. The authors put it in the title but then largely ignore it in the manuscript only mentioning it twice. Minimally you should define it precisely because of the different perspectives on what a metaorganism is. I believe that expanding on that would broaden the readership of the manuscript. I would encourage you to delve in the holobiont/metaorganism literature a little bit, and examine the papers by Seth Bordenstein and Kevin Theis (e.g., <https://journals.plos.org/plosbiology/article?id=10.1371/journal.pbio.1002226>), as well as Scott Gilbert’s on the subject. Bordenstein and Theis have worked extensively in defining the boundaries of the holobiont concept (e.g., <https://journals.asm.org/doi/full/10.1128/mSystems.00028-16>), and Gilbert tends to think about holobionts in the context of development (and by association, of regeneration) (e.g., <https://www.journals.uchicago.edu/doi/full/10.1086/668166>).

Abstract: Given that you use the term in the title, I believe you could frame your research and findings in a broader eco-evo-devo context. One sentence at the end of the abstract could possibly achieve that.

Introduction: Again in this section, mentioning 1) how common photosymbiosis is in metazoans and give a few examples; 2) how widespread regeneration is across animal phyla and better studied in some taxa vs others; and finally 3) that many photosymbiotic species regenerate pointing out how understudied that is and hence this magnificent paper. This citation may be helpful for that (<https://doi.org/10.1007/s13199-023-00920-0>). Also a couple of sentences in the life history of your acael would be useful to the non-expert.

Results: I think you do a very good job in presenting the data and taking the reader to why you zoom into runt and its role in regeneration. I would suggest you define epimorphosis and morphollaxis.

Discussion: I suggest you consider a final paragraph where you take the reader back out to thinking about animal regeneration and the role of associated microbes, and in this case photosymbionts, in this developmental process. There are many animal hosts that regenerate and they are all holobionts that we can now dissect with the very same tools you use in this paper!

Methods: Why did you feed the acoels every three days and no other feeding cycle? How did you choose the light intensity for your experiments? Was PAM performed at the same time of the light:dark cycle during the different time points? Were the acoels in the dark to measure respiration?

Spell out WISH, FISH and HCR.

Supplementary Figure 6 What are the 8 timepoints in 6a?